# Adapting to Misspecification in Contextual Bandits

**Dylan J. Foster**[⋆]
dylanf@mit.edu

**Claudio Gentile**[†]
cgentile@google.com

**Mehryar Mohri**[†‡]
mohri@google.com

**Julian Zimmert**[†]
zimmert@google.com

## Abstract

A major research direction in contextual bandits is to develop algorithms that are computationally efficient, yet support flexible, general-purpose function approximation. Algorithms based on modeling rewards have shown strong empirical performance, yet typically require a well-specified model, and can fail when this assumption does not hold. Can we design algorithms that are efficient and flexible, yet degrade gracefully in the face of model misspecification? We introduce a new family of oracle-efficient algorithms for $\varepsilon$-misspecified contextual bandits that adapt to unknown model misspecification—both for finite and infinite action settings. Given access to an *online oracle* for square loss regression, our algorithm attains optimal regret and—in particular—optimal dependence on the misspecification level, with *no prior knowledge*. Specializing to linear contextual bandits with infinite actions in $d$ dimensions, we obtain the first algorithm that achieves the optimal $\tilde{\mathcal{O}}(d\sqrt{T} + \varepsilon\sqrt{d}T)$ regret bound for unknown $\varepsilon$.

On a conceptual level, our results are enabled by a new optimization-based perspective on the regression oracle reduction framework of Foster and Rakhlin [20], which we believe will be useful more broadly.

## 1 Introduction

The contextual bandit (CB) problem is an extension of the standard multi-armed bandit problem that is relevant to a variety of applications in practice, including health services [42], online advertisement [34, 4] and recommendation systems [8]. In the contextual bandit setting, at each round, the learner observes a feature vector (or *context*) and an action set. The learner must select an action out of that set and only observes the reward of that action. To make its selection, the learner has access to a family of hypotheses (or *policies*), which map contexts to actions. The objective of the learner is to achieve a cumulative reward that is close to that of the best hypothesis in hindsight for that specific sequence of contexts and action sets.

A common approach to the contextual bandit problem consists of reducing it to a supervised learning task such as classification or regression [32, 19, 6, 7, 41, 8, 35]. Recently, Foster and Rakhlin [20] proposed SquareCB, an efficient reduction from $K$-armed contextual bandits to square loss regression under *realizability* assumptions. One open question that comes up after this work is whether their approach can be generalized to action spaces with many (or infinite) actions in $d$-dimensions. Another open question is whether one can seamlessly shift from realizability to misspecified models without requiring prior knowledge of the amount of misspecification. This is precisely the setup we study here, where the action set is large or infinite, but where the learner has a 'good' feature representation available up to some *unknown* amount of misspecification.

Adequately handling misspecification has been a subject of intense recent interest even for the simple special case of linear contextual bandits. Du et al. [18] questioned whether "good" is indeed enough,

---

[⋆]Massachusetts Institute of Technology.

[†]Google Research.

[‡]Courant Institute of Mathematical Sciences.

that is, whether we can learn efficiently even without realizability. Lattimore et al. [33] gave a positive answer to that question, provided the misspecification level $\varepsilon$ is known in advance, and showed that the price of misspecification (for regret) is roughly $\varepsilon\sqrt{dT}$, where $d$ is the dimension and $T$ is the time horizon. However, they left the adapting to unknown $\varepsilon$ as an open question.

**Our results.** We provide an affirmative answer to all of these questions. We generalize SquareCB to infinite action sets, and use this strategy to adapt to unknown misspecification $\varepsilon$ by combining it with a *bandit model selection* procedure akin to the one proposed by Agarwal et al. [9]. Our algorithm is oracle-efficient, and adapts to misspecification efficiently and optimally whenever it has access to an online oracle for square loss regression. When specialized to linear contextual bandits, it answers the question of Lattimore et al. [33].

An important conceptual contribution of our work is to show that one can view the action selection scheme used by SquareCB as an approximation to a log-barrier regularized optimization problem, which paves the way for a generalization to infinite action spaces. Another by-product of our results is a generalization of the original CORRAL algorithm [9] for combining bandit algorithms, which is simpler, flexible, and enjoys improved logarithmic factors.

## 1.1 Related Work

The contextual bandit is a well-studied problem, and misspecification in bandits and reinforcement learning has been the subject of intense recent interest. We mention a few works which are closely related to our results.

For linear bandits in $d$ dimensions, Lattimore et al. [33] gave an algorithm with regret $\mathcal{O}(d\sqrt{T} + \varepsilon\sqrt{dT})$, and left adapting to unknown misspecification for changing action sets as an open problem. Concurrent work of Pacchiano et al. [37] solves this problem for the special case where contexts/action sets are stochastic, and also leverages CORRAL-type aggregation of contextual bandit algorithms. Our results resolve this question in the general adversarial setting.

Within the literature general-purpose contextual bandit algorithms, our approach builds on a recent line of research that provides reductions to offline/online square loss regression [21, 20, 38, 45, 23].

Besides the standard references on oracle-based agnostic contextual bandits (e.g., [32, 19, 6, 7]), $\varepsilon$-misspecification is somewhat related to the recent stream of literature on bandits with adversarially-corrupted feedback [36, 26, 13]. See the discussion in the supplementary material.

## 2 Problem Setting

We consider the following contextual bandit protocol: At every round $t = 1, \ldots, T$, the learner first observes a context $x_t \in \mathcal{X}$ and an action set $\mathcal{A}_t \subseteq \mathcal{A}$, where $\mathcal{A} \subseteq \mathbb{R}^d$ is a compact action space; for simplicity, we assume throughout that $\mathcal{A} = \{a \in \mathbb{R}^d : \|a\| \leq 1\}$, but place no restriction on $(\mathcal{A}_t)_{t=1}^T$. Given the context and action set, the learner chooses action $a_t \in \mathcal{A}_t$, then observes a stochastic loss $\ell_t \in [-1, +1]$ depending on the action selected. We assume that the sequence of context vectors $x_t$ and the associated sequence of action sets $\mathcal{A}_t$ are generated by an oblivious adversary.

We let $\mu(a, x) := \mathbb{E}[\ell_t \mid x_t = x, a_t = a]$ denote the mean loss function, which is unknown to the learner. We adopt a semi-parametric approach to modeling the losses, in which $\mu(a, x)$ is modelled a (nearly) linear in the action $a$, but can depend on the context $x$ arbitrarily [20, 45, 14]. In particular, we assume the learner has access to a class of functions $\mathcal{F} \subseteq \{f : \mathcal{X} \to \mathbb{R}^d\}$, where for each $f \in \mathcal{F}$, $\langle a, f(x) \rangle$ attempts to predict the value of $\mu(a, x)$. In a well-specified/realizable setting, one would assume that there exists some $f^\star \in \mathcal{F}$ such that $\mu(a, x) = \langle a, f^\star(x) \rangle$. In this paper, we make no such assumption, but the regret incurred by our algorithms depends on how far this is from being true. For each $f \in \mathcal{F}$, we let $\pi_f(\cdot, \cdot)$ denote the *induced policy*, whose action at time $t$ is given by $\pi_f(x_t, \mathcal{A}_t) := \operatorname{argmin}_{a \in \mathcal{A}_t} \langle a, f(x_t) \rangle$.

The goal of the learner is to minimize its pseudoregret $\mathsf{Reg}(T)$ against the best unconstrained policy:

$$\mathsf{Reg}(T) := \mathbb{E}\left[\sum_{t=1}^T \mu(a_t, x_t) - \inf_{a \in \mathcal{A}_t} \mu(a, x_t)\right].$$

Here, and for the remainder of the paper, we use $\mathbb{E}[\cdot]$ to denote the expectation with respect to both the randomized choices of the learner and the stochastic realization of the losses $\ell_t$.

This setup recovers the usual finite-arm contextual bandit with $K$ arms setting by taking $\mathcal{A}_t = \{\mathbf{e}_1, \ldots, \mathbf{e}_K\}$. Another important special case is the well-studied linear contextual bandit setting, which corresponds to the case where $\mathcal{F}$ consists of constant vector-valued functions that do not depend on $\mathcal{X}$. Specifically, for any $\Theta \subseteq \mathbb{R}^d$, we can take $\mathcal{F} = \{x \mapsto \theta \,|\, \theta \in \Theta\}$. In this case, the prediction $\langle a, f(x) \rangle$ simplifies to $\langle a, \theta \rangle$, a constant linear function of the action space $\mathcal{A}$. This special case recovers the most widely studied version of linear contextual bandits [3, 12, 1, 15, 2, 10, 16], as well as Gaussian process extensions [39, 30, 17, 40].

## 2.1 Misspecification

Contextual bandit algorithms based on modeling rewards typically rely on the assumption of a *well-specified model* (or, "realizability"): That is, existence of a function $f^\star \in \mathcal{F}$ such that $\mu(a, x) = \langle a, f^\star(x) \rangle$ for all $a \in \mathcal{A}$ and $x \in \mathcal{X}$ [15, 1, 6, 21]. Since this assumption may not be realistic in practice, a more recent line of work has begun to develop algorithms for misspecified models. In particular, Crammer and Gentile [16], Ghosh et al. [25], Lattimore et al. [33] and Foster and Rakhlin [20] consider a uniform $\varepsilon$-misspecified setting in which

$$\inf_{f \in \mathcal{F}} \sup_{a \in \mathcal{A}, x \in \mathcal{X}} |\mu(a, x) - \langle a, f(x) \rangle| \leq \varepsilon, \tag{1}$$

for some misspecification level $\varepsilon > 0$. Notably, Lattimore et al. [33] show that for the linear setting, regret must grow as $\Omega(d\sqrt{T} + \varepsilon\sqrt{d}T)$. Since $d\sqrt{T}$ is the optimal regret for a well-specified model, $\varepsilon\sqrt{d}T$ may be thought of as the price of misspecification.

In this paper, we consider a weaker average-case notion of misspecification. Given a sequence $S = (x_1, \mathcal{A}_1), \ldots, (x_T, \mathcal{A}_T)$ of context-action set pairs, we define the average misspecification level $\varepsilon_T(S)$ as

$$\varepsilon_T(S) := \inf_{f \in \mathcal{F}} \left( \frac{1}{T} \sum_{t=1}^{T} \sup_{a \in \mathcal{A}_t} (\langle a, f(x_t) \rangle - \mu(a, x_t))^2 \right)^{1/2}. \tag{2}$$

This quantity measures the misspecification level for the specific sequence $S$ at hand. Of course, the uniform bound in Eq. (1) directly implies $\varepsilon_T(S) \leq \varepsilon$ for all $S$ in Eq. (2), and $\varepsilon_T(S) = 0$ whenever the model is well-specified.

We provide regret bounds that optimally adapt to $\varepsilon_T(S)$ for any given realization of the sequence $S$, with no prior knowledge of the misspecification level. The issue of adapting to unknown misspecification has not been addressed even for the stronger uniform notion (1). Indeed, previous efforts typically use prior knowledge of $\varepsilon$ to tune the exploration-exploitation scheme to encourage conservative exploration when misspecification is large; see Lattimore et al. [33, Appendix E], Foster and Rakhlin [20, Section 5.1], Crammer and Gentile [16, Section 4.2], and Zanette et al. [46] for examples. Naively adapting such schemes using, e.g., doubling tricks, presents difficulties because the quantity in Eq. (2) does not appear to be estimable without knowledge of $\mu$.

## 2.2 Regression Oracles

Following Foster and Rakhlin [20], we assume access to an *online regression oracle* SqAlg, which is simply an algorithm for sequential prediction with the square loss, using $\mathcal{F}$ as a benchmark class. More precisely, the oracle operates under the following protocol. At each round $t \in [T]$, the algorithm receives a context $x_t \in \mathcal{X}$, outputs a predictor $\hat{y}_t \in \mathbb{R}^d$ (in particular, we interpret $\langle a, \hat{y}_t \rangle$ as the predicted loss for action $a$), then observes an action $a_t \in \mathcal{A}$ and loss $\ell_t \in [-1, +1]$ and incurs loss $(\langle a_t, \hat{y}_t \rangle - \ell_t)^2$.[4] Formally, we make the following assumption.

**Assumption 1.** The regression oracle SqAlg guarantees that for any (potentially adaptively chosen) sequence $\{(x_t, a_t, \ell_t)\}_{t=1}^{T}$,

$$\sum_{t=1}^{T} (\langle a_t, \hat{y}_t \rangle - \ell_t)^2 - \inf_{f \in \mathcal{F}} \sum_{t=1}^{T} (\langle a_t, f(x_t) \rangle - \ell_t)^2 \leq \mathsf{Reg}_{\mathrm{Sq}}(T),$$

for some (non-data-dependent) upper bound $\mathsf{Reg}_{\mathrm{Sq}}(T)$.

For the finite-action setting, this definition coincides with that of Foster and Rakhlin [20]. To simplify the presentation of our results, we assume throughout the paper that $\text{Reg}_{\text{Sq}}(T)$ is a non-decreasing function of $T$.

While this type of oracle suffices for all of our results, our algorithms are stated more naturally in terms of a stronger oracle which supports *weighted* online regression. In this model, we follow the same protocol as in Assumption 1, except that at each time $t$, the regression oracle observes a weight $w_t \geq 0$ at the same time as the context $x_t$, and the loss incurred is now $w_t \cdot (\langle a_t, \hat{y}_t \rangle - \ell_t)^2$. For technical reasons, we also allow the oracle for this model to be randomized. We make the following assumption.

**Assumption 2.** The weighted regression oracle SqAlg guarantees that for any (potentially adaptively chosen) sequence $\{(w_t, x_t, a_t, \ell_t)\}_{t=1}^T$,

$$\mathbb{E}\left[\sum_{t=1}^T w_t(\langle a_t, \hat{y}_t \rangle - \ell_t)^2 - \inf_{f \in \mathcal{F}} \sum_{t=1}^T w_t(\langle a_t, f(x_t) \rangle - \ell_t)^2\right] \leq \mathbb{E}\left[\max_{t \in [T]} w_t\right] \cdot \text{Reg}_{\text{Sq}}(T),$$

for some upper bound $\text{Reg}_{\text{Sq}}(T)$, where the expectation is taken with respect to the oracle's randomization.

We show in the supplementary materialthat any unweighted regression oracle satisfying Assumption 1 can be transformed into a randomized oracle for weighted regression that satisfies Assumption 2, with no overhead in runtime. Hence, to simplify exposition, for the remainder of the paper we state our results in terms of weighted regression oracles satisfying Assumption 2.

Online regression has been well-studied, and many efficient algorithms are known for standard classes $\mathcal{F}$. One example, which is important for our applications, is when $\mathcal{F}$ is linear.

**Example 1** (Linear Models). Suppose $\mathcal{F} = \{x \mapsto \theta \mid \theta \in \Theta\}$, where $\Theta \subseteq \mathbb{R}^d$ is a convex set with $\|\theta\| \leq 1$. Then the online Newton step algorithm [27] satisfies Assumption 1 with $\text{Reg}_{\text{Sq}}(T) = \mathcal{O}(d \log(T))$ and—via our reduction —can be augmented to satisfy Assumption 2.

Further examples include kernels [44], generalized linear models [28], and standard nonparametric classes [24]. We refer to Foster and Rakhlin [20] for a more extensive discussion.

**Additional notation.** We make use of the following additional notation. Given a set $X$, we let $\Delta(X)$ denote the set of all probability distributions over $X$. If $X$ is continuous, we restrict $\Delta(X)$ to distributions with *countable* support. We let $\|x\|$ denote the euclidean norm for $x \in \mathbb{R}^d$. For any positive definite matrix $H \in \mathbb{R}^{d \times d}$, we denote the induced norm on $x \in \mathbb{R}^d$ by $\|x\|_H^2 = \langle x, Hx \rangle$. For functions $f, g : X \to \mathbb{R}_+$, we write $f = \mathcal{O}(g)$ if there exists some constant $C > 0$ such that $f(x) \leq Cg(x)$ for all $x \in X$. We write $f = \tilde{\mathcal{O}}(g)$ if $f = \mathcal{O}(g \max\{1, \text{polylog}(g)\})$, and define $\tilde{\Omega}(\cdot)$ analogously.

## 3 Adapting to Misspecification: An Oracle-Efficient Algorithm

We now present our main result: an efficient reduction from contextual bandits to online regression that adapts to unknown misspecification $\varepsilon_T(S)$ and supports infinite action sets. Our main theorem is as follows.

**Theorem 1.** *Suppose we have access to a weighted regression oracle* SqAlg *that satisfies Assumption 2 for class* $\mathcal{F}$. *Then there exists an efficient reduction which guarantees that for any sequence* $S = (x_1, \mathcal{A}_1), \ldots, (x_T, \mathcal{A}_T)$ *with misspecification level* $\varepsilon_T(S)$,

$$\text{Reg}(T) = \mathcal{O}\left(\sqrt{dT \text{Reg}_{\text{Sq}}(T) \log(T)} + \varepsilon_T(S)\sqrt{dT}\right).$$

The algorithm has building blocks: First, we extend the reduction of [20] to infinite action sets via a new optimization-based perspective, and we show that the resulting algorithm has favorable dependence on misspecification level when it is known in advance. Then, we combine this reduction with a scheme which aggregates multiple instances to adapt to unknown misspecification. If the time required for a single query to SqAlg is $\mathcal{T}_{\text{SqAlg}}$, then the per-step runtime of our algorithm is $\tilde{\mathcal{O}}(\mathcal{T}_{\text{SqAlg}} + |\mathcal{A}_t| \cdot \text{poly}(d))$.

As an important application, we solve an open question recently posed by Lattimore et al. [33]: we exhibit an efficient algorithm for infinite-action linear contextual bandits which optimally adapts to unknown misspecification.

**Corollary 1.** *Let* $\mathcal{F} = \{x \mapsto \theta \,|\, \theta \in \mathbb{R}^d, \|\theta\| \leq 1\}$. *Then there exists an efficient algorithm that, for any sequence* $S = (x_1, \mathcal{A}_1), \ldots, (x_T, \mathcal{A}_T)$, *satisfies*

$$\mathsf{Reg}(T) = \mathcal{O}\left(d\sqrt{T}\log(T) + \varepsilon_T(S)\sqrt{d}T\right).$$

This result immediately follows from Theorem 1 by applying online Newton step algorithm as the regression oracle, as in Example 1. Modulo logarithmic factors, this bound coincides with the one achieved by Lattimore et al. [33] for the simpler non-contextual linear bandit problem, for which the authors also present a matching lower bound.

The remainder of this section is dedicated to proving Theorem 1. The roadmap is as follows. First, we revisit the reduction from $K$-armed contextual bandits to online regression by Foster and Rakhlin [20] and provide a new optimization-based perspective. This new viewpoint leads to a natural generalization from the $K$-armed case to the infinite action case. We then provide an aggregation-type procedure which combines multiple instances of this algorithm to adapt to unknown misspecification, and finally put all the pieces together to prove the main result. As an extension, we also give a variant of the algorithm which enjoys improved bounds when the action sets $\mathcal{A}_t$ lie in low-dimensional subspaces of $\mathbb{R}^d$. Going forward, we abbreviate $\varepsilon_T(S)$ to $\varepsilon_T$ whenever the sequence $S$ is clear from context.

### 3.1 Oracle Reductions with Finite Actions: An Optimization-Based Perspective

An important special case of our setting, is the finite-arm contextual bandit problem, where $\mathcal{A}_t = \mathcal{K} := \{\mathbf{e}_1, \ldots, \mathbf{e}_K\}$. For this setting, Foster and Rakhlin [20] proposed an efficient and optimal reduction called SquareCB, which is displayed in Algorithm 1. At each step, queries the oracle SqAlg with the current context $x_t$ and receives a loss predictor $\hat{\theta}_t \in \mathbb{R}^K$ (so that $(\hat{\theta}_t)_i$ predicts the loss of action $i$). The algorithm then samples an action from a probability distribution introduced by Abe and Long [3]. Specifically for any $\theta \in \mathbb{R}^K$ and learning rate $\gamma > 0$, we define abe-long$(\theta, \gamma)$ as the distribution $p \in \Delta([K])$ obtained by first selecting any $i^\star \in \operatorname{argmin}_{i \in [K]} \theta_i$, then defining

$$p_i = \begin{cases} \frac{1}{K + \gamma(\theta_i - \theta_{i^\star})}, & \text{if } i \neq i^\star, \\ 1 - \sum_{i' \neq i^\star} p_i, & \text{otherwise.} \end{cases} \quad (3)$$

---
**Algorithm 1:** SquareCB [20]

**Input:** Learning rate $\gamma$, time horizon $T$.
**Initialize** Regression oracle SqAlg.
**for** $t = 1, \ldots, T$ **do**
    Receive context $x_t$.
    Let $\hat{\theta}_t$ be the oracle's prediction for $x_t$.
    Sample $I_t \sim$ abe-long$(\hat{\theta}_t, \gamma)$.
    Play $a_t = \mathbf{e}_{I_t}$ and observe loss $\ell_t$.
    Update SqAlg with $(x_t, a_t, \ell_t)$.

---

By choosing $\gamma \propto \sqrt{KT/(\mathsf{Reg}_{\mathsf{Sq}}(T) + \varepsilon_T)}$, this algorithm guarantees that

$$\mathsf{Reg}(T) \leq \mathcal{O}\left(\sqrt{KT\mathsf{Reg}_{\mathsf{Sq}}(T)} + \varepsilon_T\sqrt{KT}\right).$$

Since this approach is the starting point for our results, it will be useful to sketch the proof. For $p \in \Delta(\mathcal{A})$, let $H_p := \mathbb{E}_{a \sim p}[aa^\top]$ be the correlation matrix, and $\bar{a}_p := \mathbb{E}_{a \sim p}[a]$ be the expected action. Let the sequence $S$ be fixed, and let $f^\star \in \mathcal{F}$ be any regression function which attains the value of $\varepsilon_T(S)$ in Eq. (2).[5] With $a_t^\star := \pi_{f^\star}(x_t, \mathcal{A}_t)$ and $\theta_t^\star := f^\star(x_t)$, we have

$$\mathbb{E}\left[\sum_{t=1}^T \mu(a_t, x_t) - \inf_{a \in \mathcal{A}_t} \mu(a, x_t)\right] \leq \mathbb{E}\left[\sum_{t=1}^T \langle a_t - a_t^\star, \theta_t^\star \rangle\right] + 2\varepsilon_T T$$

$$= \mathbb{E}\left[\sum_{t=1}^T \langle \bar{a}_{p_t} - a_t^\star, \theta^\star \rangle - \frac{\gamma}{4}\|\theta^\star - \hat{\theta}_t\|_{H_{p_t}}^2\right] + \mathbb{E}\left[\sum_{t=1}^T \frac{\gamma}{4}\|\theta^\star - \hat{\theta}_t\|_{H_{p_t}}^2\right] + 2\varepsilon_T T.$$

The first expectation term above is bounded by $\mathcal{O}(KT/\gamma)$, which is established by showing that abe-long$(\hat{\theta}, \gamma)$ is an approximate solution to the per-round minimax problem

$$\min_{p \in \Delta(\mathcal{K})} \max_{\theta \in \mathbb{R}^K} \max_{a^\star \in \mathcal{K}} \langle \bar{a}_p - a^\star, \theta \rangle - \frac{\gamma}{4}\|\hat{\theta} - \theta\|_{H_p}^2, \quad (4)$$

with value $\mathcal{O}(K/\gamma)$. The second expectation term is bounded by $\mathcal{O}(\gamma \cdot (\mathsf{Reg}_{\mathrm{Sq}}(T) + \varepsilon_T T))$, which follows almost immediately from the definition of the square loss regret in Assumption 1 (see the proof of Theorem 3 for details). Choosing $\gamma$ to balance the terms leads to the result.

As a first step toward generalizing this result to infinite actions, we propose a new distribution which *exactly* solves the minimax problem (4). This distribution is the solution to a dual optimization problem based on log-barrier regularization, and provides a new principled approach to deriving reductions.

**Lemma 1.** For any $\theta \in \mathbb{R}^K$ and $\gamma > 0$, the unique minimizer of Eq. (4) coincides with the unique minimizer of the log-barrier$(\theta, \gamma)$ optimization problem defined by

$$\mathsf{log\text{-}barrier}(\theta, \gamma) = \operatorname{argmin}_{p \in \Delta([K])} \left\{ \langle p, \theta \rangle - \tfrac{1}{\gamma} \sum_{a \in [K]} \log(p_a) \right\} = \left( \tfrac{1}{\lambda + \gamma \theta_i} \right)_{i=1}^K, \qquad (5)$$

where $\lambda$ is the unique value that ensures that the weights on the right-hand side above sum to one.

The abe-long distribution is closely related to the log-barrier distribution: Rather than finding the optimal Lagrange multiplier $\lambda$ that solves the log-barrier problem, the abe-long strategy simply plugs in $\lambda = K - \gamma \min_{i'} \theta_{i'}$, then shifts weight to $p_{i^\star}$ to ensure the distribution is normalized. Since the log-barrier strategy solves the minimax problem Eq. (4) exactly, plugging it into the results of Foster and Rakhlin [20] and Simchi-Levi and Xu [38] in place of abe-long leads to slightly improved constants. More importantly, this new perspective leads to a principled way to extend these reductions to infinite actions.

### 3.2 Moving to Infinite Action Sets: The Log-Determinant Barrier

We generalize the log-barrier distribution to infinite action sets using the log-determinant function. For any $p \in \Delta(\mathcal{A})$, denote $\bar{a}_p = \mathbb{E}_{a \sim p}[a]$ and $H_p = \mathbb{E}_{a \sim p}[aa^T]$. Furthermore we use $\dim(\mathcal{A})$ to denote the dimension of the smallest affine linear subspace that contains $\mathcal{A}$. When $\dim(\mathcal{A}) < d$, we adopt the convention that $\det(\cdot)$ takes the product of only the first $\dim(\mathcal{A})$ eigenvalues of the matrix in its argument, so that the solution bwlow is well-defined. Our *logdet-barrier* distributions are defined as follows.

---
**Algorithm 2:** SquareCB.Inf
**Input:** Learning rate $\gamma$, time horizon $T$.
**Initialize** Regression oracle SqAlg.
**for** $t = 1, \ldots, T$ **do**
    Receive context $x_t$.
    Let $\hat{\theta}_t$ be the oracle's prediction for $x_t$.
    Play $a_t \sim \mathsf{logdet\text{-}barrier}(\hat{\theta}_t, \gamma; \mathcal{A}_t)$.
    Observe loss $\ell_t$.
    Update SqAlg with $(x_t, a_t, \ell_t)$.

---

**Definition 1.** For any $\theta \in \mathbb{R}^d$, action set $\mathcal{A} \subset \mathbb{R}^d$, and $\gamma > 0$, the set of logdet-barrier$(\theta, \gamma; \mathcal{A})$ distributions are defined as the solutions to

$$\operatorname{argmin}_{p \in \Delta(\mathcal{A})} \left\{ \langle \bar{a}_p, \theta \rangle - \gamma^{-1} \log \det(H_p - \bar{a}_p \bar{a}_p^T) \right\}. \qquad (6)$$

In general, Eq. (6) does not admit a unique solution; all of our results apply to *any* minimizer. Our key result is that these logdet-barrier distributions solve a minimax problem analogous to that of Eq. (4).

**Lemma 2.** Any solution to logdet-barrier$(\hat{\theta}, \gamma; \mathcal{A})$ satisfies

$$\max_{\theta \in \mathbb{R}^d} \max_{a^\star \in \mathcal{A}} \langle \bar{a}_p - a^\star, \theta \rangle - \tfrac{\gamma}{4} \|\hat{\theta} - \theta\|_{H_p}^2 \le \gamma^{-1} \dim(\mathcal{A}). \qquad (7)$$

By replacing the abe-long distribution with the logdet-barrier distribution in Algorithm 1, we obtain an optimal reduction for infinite action sets. This algorithm, which we call SquareCB.Inf, is displayed in Algorithm 2.

**Theorem 2.** *Given a regression oracle* SqAlg *that satisfies Assumption 1 for class $\mathcal{F}$,* SquareCB.Inf *with learning rate $\gamma \propto \sqrt{dT/(\mathsf{Reg}_{\mathrm{Sq}}(T) + \varepsilon)}$ guarantees for all sequences $S$ with $\varepsilon_T(S) \le \varepsilon$ that*

$$\mathsf{Reg}(T) = \mathcal{O}\left( \sqrt{dT \mathsf{Reg}_{\mathrm{Sq}}(T)} + \varepsilon \sqrt{dT} \right).$$

The logdet-barrier optimization problem is closely related to the D-optimal experimental design problem and to finding the John ellipsoid [29, 43], which correspond to the case where $\theta = 0$ in Eq. (6) [31]. By adapting specialized optimization algorithms for these problems (in particular, a Frank-Wolfe-type scheme), we can efficiently solve the logdet-barrier problem. In particular, we have the following proposition.

**Proposition 1.** *An approximation to* (6) *that achieves the same regret bound up to a constant factor can be computed in time* $\tilde{\mathcal{O}}(|\mathcal{A}_t| \cdot \mathrm{poly}(d))$ *and memory* $\tilde{\mathcal{O}}(\log|\mathcal{A}_t| \cdot \mathrm{poly}(d))$ *per round.*

The algorithm and a full analysis for runtime and memory complexity, as well as the impact on the regret, is provided in the supplementary material.

### 3.3 Adapting to Misspecification: Algorithmic Framework

The regret bound of SquareCB.Inf in Theorem 2 achieves optimal dependence on dimension and on the misspecification level, but requires an a-priori upper bound on $\varepsilon_T(S)$ to set the learning rate. We now turn our attention to adapting to this parameter.

At a high level, our approach is to run multiple instances of SquareCB.Inf, each tuned to a different level of misspecification, then run an aggregation procedure on top to learn the best instance. Specifically, if we initialize a collection of $M := \lfloor \log(T) \rfloor$ instances of Algorithm 2 in which the learning rate for instance $m$ is tuned for misspecification level $\varepsilon'_m := \exp(-m)$ (that is, we follow a geometric grid), then it is straightforward to show that there exists $m^\star \in [M]$ such that the $m^\star$th instance would enjoy optimal regret if we ran it on the sequence $S$. Since $m^\star$ is not known a-priori, we run an aggregation (or, "Corralling") procedure [9] to select the best instance. This approach is, in general, not suitable for model selection, since it typically requires prior knowledge of the optimal regret bound to tune certain parameters appropriately [22]. We show that adaptation to misspecification is an exception to this rule, and provides a simple setting where model selection for contextual bandits is possible.

We consider the aggregation scheme in Algorithm 3, which is a generalization of the CORRAL algorithm of Agarwal et al. [9]. The algorithm is initialized with $M$ *base* algorithms, and uses a multi-armed bandit algorithm with $M$ arms as a *master* algorithm responsible for choosing which base algorithm to follow at each round.

---

**Algorithm 3:** Corralling [9]

**Input:** Master algorithm Master, $T$
**Initialize** $(\mathrm{Base}_m)_{m=1}^M$
**for** $t = 1, \dots, T$ **do**
  Receive context $x_t$.
  Receive $A_t, q_{t,A_t}$ from Master.
  Pass $(x_t, \mathcal{A}_t, q_{t,A_t}, \rho_{t,A_t})$ to $\mathrm{Base}_{A_t}$.
  $\mathrm{Base}_{A_t}$ plays $a_t$ and observes $\ell_t$.
  Update Master with $\tilde{\ell}_{t,A_t} = (\ell_t + 1)$.

---

The master maintains a distribution $q_t \in \Delta([M])$ over the base algorithms. At each round $t$, it samples an algorithm $A_t \sim q_t$ and passes the current context $x_t$ into this algorithm, as well as the sampling probability $q_{t,A_t}$ and a weight $\rho_{t,A_t}$, where we define $\rho_{t,m} := 1/\min_{s \le t} q_{s,m}$ for each $m$. The base algorithm $A_t$ now plays a regular contextual bandit round: Given the context $x_t$, it proposes an arm $a_t$, which is pulled, receives the loss $\ell_t$, and updates its internal state. Finally, the master updates its state with the action-loss pair $(A_t, \tilde{\ell}_{t,A_t})$, where $\tilde{\ell}_{t,A_t} := \ell_t + 1$ (for technical reasons, it is useful to shift the loss by 1 to ensure non-negativity).

Let $\mathrm{Reg}_{\mathrm{Imp}}^m(T) := \mathbb{E}\left[\sum_{t=1}^T \frac{\mathbb{I}\{A_t = m\}}{q_{t,m}} (\mu(a_t, x_t) - \inf_{a \in \mathcal{A}_t} \mu(a, x_t))\right]$ denote the *importance-weighted regret* for base algorithm $m$, which is simply the pseudoregret incurred in the rounds where Algorithm 3 follows this base algorithm, weighted inversely proportional to the probability that this occurs. It is straightforward to show that for any choice of master and base algorithms, this scheme guarantees that

$$\mathrm{Reg}(T) = \mathbb{E}\left[\sum_{t=1}^T \tilde{\ell}_{t,A_t} - \tilde{\ell}_{t,m^\star}\right] + \mathrm{Reg}_{\mathrm{Imp}}^{m^\star}(T), \tag{8}$$

where $\tilde{\ell}_{t,m}$ henceforth denotes the loss the algorithm would have suffered at round $t$ if we had $A_t = m$. That is, the regret of Algorithm 3 is equal to the regret $\mathrm{Reg}_M(T) := \mathbb{E}[\sum_{t=1}^T \tilde{\ell}_{t,A_t} - \tilde{\ell}_{t,m^\star}]$ of the master algorithm, plus the importance-weighted regret of the optimal base algorithm $m^\star$.

The difficulty in instantiating this general scheme lies in the fact that the important-weighted regret of the best base typically scales with $\mathbb{E}[\rho_{T,m^\star}^\alpha] \cdot \mathrm{Reg}_{\mathrm{Unw}}^{m^\star}(T)$, where $\alpha \in [\frac{1}{2}, 1]$ is an algorithm-dependent parameter and $\mathrm{Reg}_{\mathrm{Unw}}^m(T) := \mathbb{E}[\sum_{t=1}^T \mathbb{I}\{A_t = m\} (\mu(a_t, x_t) - \inf_{a \in \mathcal{A}_t} \mu(a, x_t))]$ denotes the un-weighted regret of algorithm $m$. A-priori, the $\mathbb{E}[\rho_{T,m^\star}^\alpha]$ can be unbounded, leading to large regret. The key to the analysis of Agarwal et al. [9], and the approach we follow here, is to use a master algorithm with *negative regret* proportional to $\mathbb{E}[\rho_{T,m^\star}^\alpha]$, allowing to cancel this factor.

**Base algorithm.** As the first step towards instantiating the aggregation scheme above, we specify the base algorithm. We use a modification to SquareCB.Inf based importance weighting, which is designed to ensure that the importance-weighted regret in Eq. (8) is bounded. Pseudocode for the $m$th base algorithm is given in Algorithm 4.

Let the instance $m$ be fixed, and let $Z_{t,m} = \mathbb{I}\{A_t = m\}$ indicate the event that this instance gets to select an arm; note that we have $Z_{t,m} \sim \text{Ber}(q_{t,m})$ marginally. When $Z_{t,m} = 1$, instance $m$ receives $q_{t,m}$ and $\rho_{t,m} = \max_{s \leq t} q_{s,m}^{-1}$ from the master algorithm. The

---

**Algorithm 4:** SquareCB.Imp (for base alg. $m$)

**Input:** $T$, $\text{Reg}_{\text{Sq}}(T)$
**Initialize** Weighted regression oracle SqAlg.
**for** $t = (\tau_1, \tau_2, \ldots) \subset [T]$ **do**
  Receive context $x_t$ and $(q_{t,m}, \rho_{t,m})$.
  Set $\gamma_{t,m} =$
   $\min\left\{ \frac{\sqrt{d}}{\varepsilon'_m}, \sqrt{dT/(\rho_{t,m}\text{Reg}_{\text{Sq}}(T))} \right\}$.
  Set $w_t = \gamma_{t,m}/q_{t,m}$.
  Compute oracle's prediction $\hat{\theta}_t$ for $x_t, w_t$.
  Sample $a_t \sim \text{logdet-barrier}(\theta_t, \gamma_{t,m}; \mathcal{A}_t)$.
  Play $a_t$ and observe loss $\ell_t$.
  Update SqAlg with $(w_t, x_t, a_t, \ell_t)$.

---

instance then follows the same update scheme as in the vanilla version of SquareCB.Inf, except that i) it uses an adaptive learning rate $\gamma_{t,m}$, which is tuned based on $\rho_{t,m}$, and ii) it uses a weighted square loss regression oracle (Assumption 2), with the weight $w_t$ set as a function of $\gamma_{t,m}$ and $q_{t,m}$.

The importance weighted regret $\text{Reg}^m_{\text{Imp}}(T)$ for this scheme is bounded as follows.

**Theorem 3.** *When invoked within Algorithm 3 with a regression oracle satisfying Assumption 2, the importance-weighted regret for each instance $m \in [M]$ of Algorithm 4 satisfies*

$$\text{Reg}^m_{\text{Imp}}(T) \leq \tfrac{3}{2}\mathbb{E}[\sqrt{\rho_{T,m}}]\sqrt{dT\text{Reg}_{\text{Sq}}(T)} + \left(\left(\tfrac{\varepsilon'_m}{\varepsilon_T} + \tfrac{\varepsilon_T}{\varepsilon'_m}\right)\sqrt{d} + 2\right)\varepsilon_T T. \tag{9}$$

The key feature of this regret bound is that only the leading term involving $\text{Reg}_{\text{Sq}}(T)$ depends on the importance weights, not the second term involving the misspecification. This is allows us to get away with tuning the master algorithm using only $d$, $T$, and $\text{Reg}_{\text{Sq}}(T)$, but not $\varepsilon_T$, which is critical to adapt without prior knowledge. Another important detail is that if $\varepsilon'_m$ is within a constant factor of $\varepsilon_T$, the second term simplifies to $\mathcal{O}(\varepsilon_T\sqrt{dT})$ as desired.

## 3.4 Improved Master Algorithms for Combining Bandit Algorithms

It remains to provide a master algorithm for use within Algorithm 3. While it turns out the master algorithm proposed in Agarwal et al. [9] suffices for this task, we go a step further and propose a new master algorithm called $(\alpha, R)$–*hedged FTRL* which is simpler and enjoys slightly improved regret, removing logarithmic factors. While this is not the focus of the paper, we believe that it to be a useful contribution on its own, because it provides a new approach to designing master algorithms for bandit aggregation. We hope that it will find use more broadly.

The $(\alpha, R)$–hedged FTRL algorithm is parameterized by a regularizer and two scale parameters $\alpha \in (0,1)$ and $R > 0$. We defer a precise definition and analysis to supplementary material, and state only the relevant result for our aggregation setup here. This result concerns a specific instance of the $(\alpha, R)$–hedged FTRL algorithm that we call $(\alpha, R)$–hedged Tsallis-INF, which instantiates the framework using the Tsallis entropy as a regularizer [11, 5, 47]. The key property of the algorithm is that the regret with respect to a policy playing a fixed arm $m$ contains a negative contribution of magnitude $\rho^\alpha_{T,m}R$. The following result is a corollary of a more general theorem, found in the supplementary material.

**Corollary 2.** *Consider the adversarial multi-armed bandit problem with $M$ arms and losses $\tilde{\ell}_{t,m} \in [0,2]$. For any $\alpha \in (0,1)$ and $R > 0$, the $(\alpha, R)$–hedged Tsallis-INF algorithm with learning rate $\eta = \sqrt{1/(2T)}$ guarantees that for all $m^\star \in [M]$,*

$$\mathbb{E}\left[\sum_{t=1}^T \tilde{\ell}_{t,A_t} - \tilde{\ell}_{t,m^\star}\right] \leq 4\sqrt{2MT} + \mathbb{E}\left[\min\left\{\tfrac{1}{1-\alpha}, 2\log(\rho_{T,m^\star})\right\}M^\alpha - \rho^\alpha_{T,m^\star}\right] \cdot R. \tag{10}$$

## 3.5 Putting Everything Together

Crucially, the regret bound in Corollary 2 has a negative $R \cdot \rho^\alpha_{T,m^\star}$ term which, for sufficiently large $R$ and appropriate $\alpha$, can be used to offset the regret incurred from importance-weighting the base

algorithms. In particular, $\left(\frac{1}{2}, \frac{3}{2}\sqrt{dT\mathsf{Reg}_{\mathrm{Sq}}(T)}\right)$–hedged Tsallis-INF has exactly the negative regret contribution needed to cancel the importance weighting term in Eq. (9) if we use SquareCB.Imp as the base algorithm. In more detail, we combine the regret for the master and base algorithms as follows to prove Theorem 1.

**Proof sketch for Theorem 1.** Using Eq. (8), it suffices to bound the regret of the bandit master $\mathsf{Reg}_M(T)$ and the important-weighted regret $\mathsf{Reg}_{\mathrm{Imp}}^{m^\star}(T)$ for the optimal instance $m^\star$. By Corollary 2, using $\left(\frac{1}{2}, \frac{3}{2}\sqrt{dT\mathsf{Reg}_{\mathrm{Sq}}(T)}\right)$–hedged Tsallis-INF as the master algorithm gives

$$\mathsf{Reg}_M(T) \leq \mathcal{O}\left(\sqrt{dT\mathsf{Reg}_{\mathrm{Sq}}(T)\log(T)}\right) - \tfrac{3}{2}\mathbb{E}[\sqrt{\rho_{T,m^\star}}]\sqrt{dT\mathsf{Reg}_{\mathrm{Sq}}(T)}.$$

Whenever the misspecification level is not trivially small, the geometric grid ensures that there exists $m^\star \in [M]$ such that $e^{-1}\varepsilon_T \leq \varepsilon'_{m^\star} \leq \varepsilon_T$. For this instance, Theorem 3 yields

$$\mathsf{Reg}_{\mathrm{Imp}}^{m^\star}(T) \leq \tfrac{3}{2}\mathbb{E}[\sqrt{\rho_{T,m^\star}}]\sqrt{dT\mathsf{Reg}_{\mathrm{Sq}}(T)} + \mathcal{O}(\varepsilon_T\sqrt{dT}).$$

Summing the two bounds using Eq. (8) completes the proof. $\qquad\square$

### 3.6 Extension: Adapting to the Average Dimension

A canonical application of linear contextual bandit is the problem of online news article recommendation, where the context $x_t$ is taken to be a feature vector containing information about the user, and each action $a \in \mathcal{A}_t$ is the concatenation of $x_t$ with a feature representation for a candidate article (e.g., Li et al. [34]). In this application and others like it, it is often the case that while examples lie in high-dimensional space, the true dimensionality $\dim(\mathcal{A}_t)$ of the action set is small, so that $d_{\mathrm{avg}} := \frac{1}{T}\sum_{t=1}^T \dim(\mathcal{A}_t) \ll d$. If we have prior knowledge of $d_{\mathrm{avg}}$ (or an upper bound thereof), we can exploit this low dimensionality for tighter regret. In fact, following the proof of Theorem 3 and Theorem 1, and bounding $\sum_{t=1}^T \dim(A_t)$ by $d_{\mathrm{avg}}T$ instead of $dT$, it is fairly immediate to show that Algorithm 3 enjoys improved regret $\mathsf{Reg}(T) = \mathcal{O}(\sqrt{d_{\mathrm{avg}}T\mathsf{Reg}_{\mathrm{Sq}}(T)\log(T)} + \varepsilon_T\sqrt{d_{\mathrm{avg}}T})$, so long as $d_{\mathrm{avg}}$ is replaced by $d$ in the algorithm's various parameter settings. Our final result shows that it is possible to adapt to unknown $d_{\mathrm{avg}}$ and unknown misspecification simultaneously. The key idea to apply a doubling trick on top of Algorithm 3

**Theorem 4.** *There exists an algorithm that, under the same conditions as Theorem 1, satisfies* $\mathsf{Reg}(T) = \mathcal{O}\left(\sqrt{d_{\mathrm{avg}}T\mathsf{Reg}_{\mathrm{Sq}}(T)\log(T)} + \varepsilon_T\sqrt{d_{\mathrm{avg}}T}\right)$ *without prior knowledge of $d_{\mathrm{avg}}$ or $\varepsilon_T$.*

We remark that while the bound in Theorem 4 replaces the $d$ factor in the reduction with the data-dependent quantity $d_{\mathrm{avg}}$, the oracle's regret $\mathsf{Reg}_{\mathrm{Sq}}(T)$ may itself still depend on $d$ unless a sufficiently sophisticated algorithm is used.

## 4 Discussion

We have presented the first general-purpose, oracle-efficient algorithms for contextual bandits that adapt to unknown model misspecification. For infinite-action linear contextual bandits, our results yield the first optimal algorithms that adapt to unknown misspecification with changing action sets. Our results suggest a number of interesting conceptual questions:

- Can our optimization-based perspective lead to new oracle-based algorithms for more rich types of infinite action sets? Examples include nonparametric actions and structured (e.g., sparse) linear actions.
- Can our reduction-based techniques be lifted to more sophisticated interactive learning settings such as reinforcement learning?

On the technical side, we anticipate that our new approach to reductions will find broader use; natural extensions include reductions for offline oracles [38] and adapting to low-noise conditions [23].

Lastly, we recall that in passing, we have derived a novel class of master algorithms for combining bandit algorithms which enjoys more flexibility, an improvement in logarithmic factors, and a greatly simplified analysis. We hope this result will be useful for future work on model selection in contextual bandits.

**Acknowledgements**

DF acknowledges the support of NSF TRIPODS grant #1740751. We thank Teodor Marinov and Alexander Rakhlin for discussions on related topics.

## Broader Impact

This paper concerns contextual bandit algorithms that adapt to unknown model misspecification. Because of their efficiency and ability to adapt to the amount of misspecification contained with no prior knowledge, our algorithms are robust, and may be suitable for large-scale practical deployment. On the other hand, our work is at the level of foundational research, and hence its impact on society is shaped by the applications that stem from it. We will focus our brief discussion on the applications mentioned in the introduction.

Health services [42] offer an opportunity for potential positive impact. Contextual bandits can be used to propose medical interventions that lead to a better health outcomes. However, care must be taken to ethically implement the explore-exploit tradeoff in this sensitive setting, and more research is required. Online advertisements [4, 34] and recommendation systems [8] are another well-known application. While improved, robust algorithms can lead to increased profits here, it is important to recognize that this may positively impact society as a whole.

Lastly, we mention that predictive algorithms like contextual bandits become more and more powerful as more information is gathered about users. This provides a clear incentive toward collecting as much information as possible. We believe that the net benefit of research on contextual bandit outweighs the harm, but we welcome regulatory efforts to produce a legal framework that steers the usage of machine learning algorithms, including in contextual bandits, in a direction which is respects of the privacy rights of users.

## Footnotes

[4]As in Foster and Rakhlin [20], the *square loss* itself does not play a crucial role, and can be replaced by other loss that is strongly convex with respect to the learner's predictions.

[5]If the infimum is not obtained, we can simply apply the argument that follows with a limit sequence.

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
