[Supplementary Material]

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

[6]Note that the use of the log-barrier in CORRAL is not related to our use of the log-barrier within the contextual bandit framework.

[7]For time-dependent learning rates, replace $\eta$ by $\eta_t$ in the update rule of Eq. (18).

[8]Abernethy et al. [5] present this result slightly differently. See our proof of Theorem 6 with $R = 0$ for an alternative.

[9]This has been shown for $L = 1$ but the extension to general $L$ is trivial.

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

---

**Algorithm 5:** Randomized reduction from weighted to unweighted online regression

---

**Input:** Online regression oracle SqAlg satisfying Assumption 1.
**Initialize** $w_{\max} \leftarrow 0$
**for** $t = 1, \ldots, T$ **do**

> Receive weight $w_t$ and $x_t$.
> **if** $w_t > w_{\max}$ **then**
>> Reset SqAlg.
>> $w_{\max} \leftarrow 2w_t$.
>
> Predict $\hat{y}_t$, where $\hat{y}_t$ is the prediction from SqAlg given $x_t$.
> Observe $a_t$ and $\ell_t$.
> **if** $u_t \sim \mathrm{Ber}(w_t/w_{\max}) = 1$ **then**
>> Update SqAlg with $(x_t, a_t, \ell_t)$.

---

## A  Additional Related Work

In particular, our work builds on and provides a new perspective on the online square loss oracle reduction of Foster and Rakhlin [21]. The infinite-action setting we consider was introduced in Foster and Rakhlin [21], but algorithms were only given for the special case where the action set is the sphere; our work extends this to arbitrary action sets. Concurrent work of Xu and Zeevi [46] gives a reduction to offline oracles for infinite action sets. This result is not strictly comparable: On one hand, an online oracle can always be converted to an offline oracle through online-to-batch conversion, but when an online oracle *is* available our algorithm is significantly more efficient.

Misspecification in contextual bandits can be formalized in different ways that go beyond the setting we consider. First, we mention a long line of work which reduces stochastic contextual bandits to oracles for cost-sensitive classification [33, 20, 6, 7]. These results are agnostic, meaning they make no assumption on the model. However, in the presence of misspecification, the type of guarantee is somewhat different than what we provide here: rather than giving a bound on regret to the true optimal policy, these results give bounds on the regret to the best-in-class policy.

Another line of works consider a model in which the feedback received by the learning algorithm at each round may be arbitrarily corrupted by an adaptive adversary [37, 27, 14]. Typical results for this setting pick up additive error $\mathcal{O}(C)$, where $C$ is the total number of corrupted rounds. While this model was original introduced for non-contextual stochastic bandits, it has recently been extended to Gaussian process bandit optimization, which is closely related to the contextual bandit setting (though these results only tolerate $C \le \sqrt{T}$). While this is not the focus of our paper, we mention in passing that our notion of misspecification satisfies $\varepsilon_T(S) \le \sqrt{C/T}$, and hence our main theorem (Theorem 1) picks up additive error $\sqrt{CT}$ for this corrupted setting (albeit, only with an oblivious adversary).

## B  Reducing Weighted to Unweighted Regression

In this section we show how to transform any unweighted online regression oracle SqAlg satisfying Assumption 1 into a weighted oracle satisfying Assumption 2. The reduction is stated in Algorithm 5.

**Theorem 5.** *If the oracle* SqAlg *satisfies Assumption 1 with regret bound* $\mathrm{Reg}_{\mathrm{Sq}}(T)$*, then Algorithm 5 satisfies Assumption 2 with regret bound* $\mathrm{Reg}_{\mathrm{Sq}}(T)$*.*

**Proof.** Let $D_t = (w_t, x_t, a_t, \ell_t)$ and define the filtration $\mathfrak{F}_t = \sigma(D_{1:t})$, with the convention $\mathbb{E}_t[\cdot] = \mathbb{E}[\cdot \mid \mathfrak{F}_t]$. Let $\tau_1, \tau_2 \ldots, \tau_I$ denote the timesteps at which the algorithm doubles $w_{\max}$ and resets SqAlg, with the convention $\forall n > I : \tau_n = T + 1$. Note that these random variables are stopping times with respect to the filtration $\mathfrak{F}_{1:T}$, and hence $\mathfrak{F}_{\tau_i}$ is well-defined for each $i \in \mathbb{N}$. It will also be helpful to note that we always have $\tau_{i+1} > \tau_i$ for all $i \le I$ by construction and otherwise $\tau_{i+1} = \tau_i$. We also observe that $\tau_1 = 1$ unless $w_1 = 0$.

For the first step, we show that the conditional regret of Algorithm 5 between any pair of doubling steps is bounded. Let $i \le I$ and $f \in \mathcal{F}$ be fixed, and observe that $i \le I$ holds iff $\tau_i \le T$, which is

$\mathfrak{F}_{\tau_i}$-measurable. Hence,

$$\mathbb{E}\left[\sum_{t=\tau_i}^{\tau_{i+1}-1} w_t\big((\langle a_t, \hat{y}_t\rangle - \ell_t)^2 - (\langle a_t, f(x_t)\rangle - \ell_t)^2\big) \mid \mathfrak{F}_{\tau_i}\right]$$

$$= \mathbb{E}\left[2w_{\tau_i}\sum_{t=\tau_i}^{\tau_{i+1}-1} \frac{w_t}{2w_{\tau_i}}\big((\langle a_t, \hat{y}_t\rangle - \ell_t)^2 - (\langle a_t, f(x_t)\rangle - \ell_t)^2\big) \mid \mathfrak{F}_{\tau_i}\right]$$

$$\overset{(a)}{=} \mathbb{E}\left[2w_{\tau_i}\sum_{t=\tau_i}^{\tau_{i+1}-1} \mathbb{E}_t\big[u_t\big((\langle a_t, \hat{y}_t\rangle - \ell_t)^2 - (\langle a_t, f(x_t)\rangle - \ell_t)^2\big)\big] \mid \mathfrak{F}_{\tau_i}\right]$$

$$\overset{(b)}{=} \mathbb{E}\left[2w_{\tau_i}\sum_{t=\tau_i}^{\tau_{i+1}-1} u_t\big((\langle a_t, \hat{y}_t\rangle - \ell_t)^2 - (\langle a_t, f(x_t)\rangle - \ell_t)^2\big) \mid \mathfrak{F}_{\tau_i}\right]$$

$$\overset{(c)}{\leq} \mathbb{E}[2w_{\tau_i} \mid \mathfrak{F}_{\tau_i}] \cdot \mathsf{Reg}_{\mathrm{Sq}}(T)\,,$$

where (a) follows from the conditional independence of $u_t$, (b) is by the tower rule of expectation, and (c) uses Assumption 1 on the set $\{t \in \{\tau_i, \dots \tau_{i+1} - 1\} \mid u_t = 1\}$ (in particular, that regret is bounded by $\mathsf{Reg}_{\mathrm{Sq}}(T)$ on every sequence with probability 1 and $\mathsf{Reg}_{\mathrm{Sq}}(T)$ is non-decreasing in $T$). For $i > I$, the term is 0 since the sum is empty. To complete the proof that Algorithm 5 satisfies Assumption 2, we sum the bound above across all epochs as follows:

$$\mathbb{E}\left[\sum_{t=1}^{T} w_t\big((\langle a_t, \hat{y}_t\rangle - \ell_t)^2 - (\langle a_t, f(x_t)\rangle - \ell_t)^2\big)\right]$$

$$\overset{(d)}{=} \mathbb{E}\left[\sum_{i=1}^{\infty}\sum_{t=\tau_i}^{\tau_{i+1}-1} w_t\big((\langle a_t, \hat{y}_t\rangle - \ell_t)^2 - (\langle a_t, f(x_t)\rangle - \ell_t)^2\big)\right]$$

$$\overset{(e)}{=} \mathbb{E}\left[\sum_{i=1}^{\infty} \mathbb{E}\left[\sum_{t=\tau_i}^{\tau_{i+1}-1} w_t\big((\langle a_t, \hat{y}_t\rangle - \ell_t)^2 - (\langle a_t, f(x_t)\rangle - \ell_t)^2\big) \mid \mathfrak{F}_{\tau_i}\right]\right]$$

$$\overset{(f)}{\leq} \mathbb{E}\left[\sum_{i=1}^{I} \mathbb{E}[2w_{\tau_i} \mid \mathfrak{F}_{\tau_i}]\right]\mathsf{Reg}_{\mathrm{Sq}}(T)$$

$$\overset{(g)}{=} 2\mathbb{E}\left[\sum_{i=0}^{I} w_{\tau_i}\right]\mathsf{Reg}_{\mathrm{Sq}}(T)$$

$$\overset{(h)}{\leq} 2\mathbb{E}[2w_{\tau_I}]\mathsf{Reg}_{\mathrm{Sq}}(T) \overset{(i)}{\leq} 4\mathbb{E}\left[\max_{t\in[T]} w_t\right]\mathsf{Reg}_{\mathrm{Sq}}(T)\,,$$

where (d) uses that all $t < \tau_1$ have $w_t = 0$, (e) uses the tower rule of expectation, (f) applies the conditional bound between stopping times above, (g) uses the tower rule of expectation again, (h) holds because the weights at least double between doubling steps, and (i) follows because $\tau_I$ is a random variable with support over $[T]$. $\qquad\square$

## C Proofs from Section 3

In this section we provide complete proofs for all of the algorithmic results from Section 3.

### C.1 Proofs from Section 3.1

**Proof of Lemma 1.** We begin by showing that the log-barrier distribution takes the form claimed in Eq. (5). The minimization problem of Lemma 1 is strictly convex and the value is $\infty$ on the boundary. Hence the unique solution lies in the interior of the domain. By the K.K.T. conditions, the partial

derivatives in each coordinate must coincide for the minimizer $p^\star$. There exists a $\tilde{\lambda} \in \mathbb{R}$ such that

$$\forall a \in [K] : \frac{\partial}{\partial p_a} \left( \langle p^\star, \theta \rangle - \frac{1}{\gamma} \sum_{a \in [K]} \log(p_a^\star) \right) = \theta_a - \frac{1}{\gamma p_a^\star} = \tilde{\lambda} \,.$$

Substituting $\tilde{\lambda} = \min_{a \in [K]} \theta_a - 1/\gamma$ and rearranging finishes the proof.

Next we show that the log-barrier distribution indeed solves the minimax problem Eq. (4), which we rewrite as

$$\min_{p \in \Delta([K])} \sup_{\theta \in \mathbb{R}^K} \max_{i^\star \in [K]} \langle \bar{a}_p - \mathbf{e}_{i^\star}, \theta \rangle - \frac{\gamma}{4} \|\hat{\theta} - \theta\|_{H_p}^2$$

$$= \min_{p \in \Delta([K])} \max_{i^\star \in [K]} \sup_{\delta \in \mathbb{R}^K} \langle \bar{a}_p - \mathbf{e}_{i^\star}, \hat{\theta} + \delta \rangle - \frac{\gamma}{4} \|\delta\|_{H_p}^2 \,. \tag{11}$$

For any choice of $p$ and $i^\star$, taking the derivative of the expression in Eq. (11) with respect to $\delta$, we have

$$\frac{\partial}{\partial \delta} \left[ \langle \bar{a}_p - \mathbf{e}_{i^\star}, \delta \rangle - \frac{\gamma}{4} \|\delta\|_{H_p}^2 \right] = \bar{a}_p - \mathbf{e}_{i^\star} - \frac{\gamma}{2} H_p \delta \,. \tag{12}$$

For $p$ on the boundary of $\Delta([K])$ (i.e. there exists $i \in [K]$ such that $p_i = 0$), the gradient is constant and the supremum has value $+\infty$. Hence, we only need to consider the case where $p$ lies in the interior of $\Delta([K])$, which implies $H_p \succ 0$. In this case Eq. (12) is strongly convex in $\delta$ and the unique maximizer is given by $\delta^\star = \frac{2}{\gamma} H_p^{-1}(\bar{a}_p - \mathbf{e}_{i^\star})$. Hence, we can rewrite (11) as

$$\min_{p \in \Delta([K])} \max_{i^\star \in [K]} \max_{\delta \in \mathbb{R}^K} \langle \bar{a}_p - \mathbf{e}_{i^\star}, \hat{\theta} + \delta \rangle - \frac{\gamma}{4} \|\delta\|_{H_p}^2$$

$$= \min_{\substack{p \in \Delta([K]) \\ H_p \succ 0}} \max_{i^\star \in [K]} \langle \bar{a}_p - \mathbf{e}_{i^\star}, \hat{\theta} \rangle + \frac{1}{\gamma} \|\bar{a}_p - \mathbf{e}_{i^\star}\|_{H_p^{-1}}^2$$

$$\geq \min_{\substack{p \in \Delta([K]) \\ H_p \succ 0}} \mathbb{E}_{i^\star \sim p} \left[ \langle \bar{a}_p - \mathbf{e}_{i^\star}, \hat{\theta} \rangle + \frac{1}{\gamma} \|\bar{a}_p - \mathbf{e}_{i^\star}\|_{H_p^{-1}}^2 \right] \tag{13}$$

$$= \min_{\substack{p \in \Delta([K]) \\ H_p \succ 0}} \mathbb{E}_{i^\star \sim p} \left[ \frac{1}{\gamma} \left( \mathrm{tr}(H_p H_p^{-1}) - \|\bar{a}_p\|_{H_p^{-1}}^2 \right) \right] = \frac{K-1}{\gamma} \,.$$

Now consider the inequality (13). If we can show that there exists a unique solution $p$ such that this step in fact holds with equality, then we have identified the minimizer over $p \in \Delta([K])$. Consider an arbitrary candidate solution $p$ on the interior of $\Delta([K])$. Then, letting $W_i := \langle \bar{a}_p - \mathbf{e}_{i^\star}, \hat{\theta} \rangle + \frac{1}{\gamma} \|\bar{a}_p - \mathbf{e}_{i^\star}\|_{H_p^{-1}}^2$, the step (13) lower bounds $\max_{i \in [K]} W_i$ by $\mathbb{E}_{i \sim p}[W_i]$. This step holds with equality if and only if $\mathbb{E}_{i \sim p}[W_i - \max_{i' \in [K]} W_{i'}] = 0$. Since all probabilities $p_i$ are strictly positive, this can happen if and only if

$$\exists \tilde{\lambda} \in \mathbb{R} \quad \text{such that} \quad \forall i \in [K] : W_i = \langle \bar{a}_p - \mathbf{e}_i, \hat{\theta} \rangle + \frac{1}{\gamma} \|\bar{a}_p - \mathbf{e}_i\|_{H_p^{-1}}^2 = \tilde{\lambda} \,.$$

Basic algebra shows that

$$\langle \bar{a}_p - \mathbf{e}_i, \hat{\theta} \rangle + \frac{1}{\gamma} \|\bar{a}_p - \mathbf{e}_{i^\star}\|_{H_p^{-1}}^2 = \sum_{i' \in [K]} p_{i'} \hat{\theta}_i - \hat{\theta}_i - \frac{1}{\gamma} + \frac{1}{\gamma p_i} = \tilde{\lambda} \,.$$

Substituting $\tilde{\lambda} = \sum_{i' \in [K]} p_{i'} \hat{\theta}_i - \min_j \hat{\theta}_j - \frac{1}{\gamma} + \lambda \gamma$, rearranging and picking the unique value such that this is a probability distribution leads to the log-barrier distribution. $\qquad \square$

## C.2 Proofs from Section 3.2

Recall that $\dim(\mathcal{A})$ is the dimension of the smallest affine linear subspace containing $\mathcal{A}$. In other words $\forall a \in \mathcal{A} : \dim(\mathcal{A}) = \dim(\mathrm{span}(\mathcal{A} - a))$. Our main result in this section is the following slightly stronger version of Lemma 2.

**Lemma 3.** Any solution $p \in \Delta(\mathcal{A})$ to the problem logdet-barrier$(\hat{\theta}, \gamma; \mathcal{A})$ in Eq. (6) satisfies

$$\max_{a^\star \in \mathcal{A}} \sup_{\theta \in \mathbb{R}^d} \langle \bar{a}_p - a^\star, \theta \rangle - \frac{\gamma}{4} \|\hat{\theta} - \theta\|^2_{H_p - \bar{a}_p \bar{a}_p^\top} \leq \gamma^{-1} \dim(\mathcal{A}).$$

Since $-\|\hat{\theta} - \theta\|^2_{H_p - \bar{a}_p \bar{a}_p^\top} = -\|\hat{\theta} - \theta\|^2_{H_p} + \langle \hat{\theta} - \theta, \bar{a}_p \rangle^2 \geq -\|\hat{\theta} - \theta\|^2_{H_p}$, Lemma 2 is a direct corollary of Lemma 3.

Before proving Lemma 3, we discuss in detail the case where $\mathcal{A}$ does not span $\mathbb{R}^d$.

### C.2.1 Handling the case where $\dim(\mathcal{A}) < d$.

We first show that if $\dim(\mathcal{A}) < d$, there exists a bijection of $\mathcal{A}$ to a set $\tilde{\mathcal{A}} \subset \mathbb{R}^{\dim(\mathcal{A})}$ and a projection $P$ of the loss estimator into $\mathbb{R}^{\dim(\mathcal{A})}$, such that logdet-barrier$(\theta, \gamma; \mathcal{A})$ and logdet-barrier$(P(\theta), \gamma; \tilde{\mathcal{A}})$ are (up to the bijection) identical, and such that the objective in Lemma 3 coincides. This implies for all subsequent sections, we can assume w.l.o.g. that $\dim(\mathcal{A}) = d$, since if this does not hold we can work in the subspace outlined in this section.

We pick an arbitrary anchor $\mathfrak{a} \in \mathcal{A}$, let $P$ be the projection onto $\text{span}(\mathcal{A} - \mathfrak{a})$ represented in a fixed arbitrary orthonormal basis of $\text{span}(\mathcal{A} - \mathfrak{a})$. Denote $\tilde{\mathcal{A}} = P(\mathcal{A} - \mathfrak{a})$ and for $p \in \Delta(\mathcal{A})$ let $\tilde{p} \in \Delta(\tilde{\mathcal{A}})$ be such that $\tilde{p}_{P(a-\mathfrak{a})} = p_a$ (recall that we define $\Delta(\mathcal{A})$ to have countable support). Let $\hat{\theta} \in \mathbb{R}^d$ be arbitrary, then

$$\langle \bar{a}_p, \hat{\theta} \rangle = \mathbb{E}_{a \sim p}\Big[\langle P(a - \mathfrak{a}), P(\hat{\theta}) \rangle\Big] + \langle \mathfrak{a}, \hat{\theta} \rangle = \langle \bar{a}_{\tilde{p}}, P(\hat{\theta}) \rangle + \langle \mathfrak{a}, \hat{\theta} \rangle.$$

Recall that we define the $\det$ in logdet-barrier as the product over the first $\dim(\mathcal{A})$ eigenvalues of $H_p - \bar{a}_p \bar{a}_p^\top$. Let $(\nu_i)_{i=1}^{\dim(\mathcal{A})}$ denote the corresponding eigenvectors (note that this requires $\nu_i \in \text{span}(\mathcal{A} - \mathfrak{a})$). We have

$$\log\det(H_p - \bar{a}_p \bar{a}_p^\top) = \sum_{i=1}^{\dim(\mathcal{A})} \log(\|\nu_i\|^2_{H_p - \bar{a}_p \bar{a}_p^\top}) = \sum_{i=1}^{\dim(\mathcal{A})} \log(\mathbb{E}_{a \sim p}[\langle \nu_i, a - \bar{a}_p \rangle^2])$$

$$= \sum_{i=1}^{\dim(\mathcal{A})} \log(\mathbb{E}_{a \sim p}[\langle \nu_i, a - \mathfrak{a} - \mathbb{E}_{a' \sim p}(a' - \mathfrak{a}) \rangle^2]) = \sum_{i=1}^{\dim(\mathcal{A})} \log(\mathbb{E}_{a \sim p}[\langle P(\nu_i), P(a - \mathfrak{a}) - \bar{a}_{\tilde{p}} \rangle^2])$$

$$= \sum_{i=1}^{\dim(\mathcal{A})} \log(\|P(\nu_i)\|^2_{H_{\tilde{p}} - \bar{a}_{\tilde{p}} \bar{a}_{\tilde{p}}^\top}) = \log\det(H_{\tilde{p}} - \bar{a}_{\tilde{p}} \bar{a}_{\tilde{p}}^\top),$$

where we use the fact that $P$ only changes the representation on $\text{span}(\mathcal{A} - \mathfrak{a})$ and does not change the identity of the eigenvalues. Combining these two results immediately shows that for any $p \in$ logdet-barrier$(\hat{\theta}, \gamma; \mathcal{A})$ it follows that $\tilde{p} \in$ logdet-barrier$(P(\hat{\theta}), \gamma; \tilde{\mathcal{A}})$ and vice versa.

For the objective of Lemma 3, we have

$$\langle \bar{a}_p - a^\star, \theta \rangle = \langle \mathbb{E}_{a \sim p}[P(a - \mathfrak{a})] - P(a^\star - \mathfrak{a}), P(\theta) \rangle = \langle \bar{a}_{\tilde{p}} - (P(a^\star - \mathfrak{a})), P(\theta) \rangle.$$

For the quadratic term, following the same steps as above for $\nu_i$, we have

$$\|\hat{\theta} - \theta\|^2_{H_p - \bar{a}_p \bar{a}_p^\top} = \|P(\hat{\theta}) - P(\theta)\|^2_{H_{\tilde{p}} - \bar{a}_{\tilde{p}} \bar{a}_{\tilde{p}}^\top}.$$

and

$$\langle \bar{a}_p - a^\star, \theta \rangle - \frac{\gamma}{4} \|\hat{\theta} - \theta\|^2_{H_p - \bar{a}_p \bar{a}_p^\top} = \langle \bar{a}_{\tilde{p}} - P(a^\star - \mathfrak{a}), P(\theta) \rangle - \frac{\gamma}{4} \|P(\hat{\theta}) - P(\theta)\|^2_{H_{\tilde{p}} - \bar{a}_{\tilde{p}} \bar{a}_{\tilde{p}}^\top}.$$

Hence, we have

$$\max_{a^\star \in \mathcal{A}} \sup_{\theta \in \mathbb{R}^d} \langle \bar{a}_p - a^\star, \theta \rangle - \frac{\gamma}{4} \|\hat{\theta} - \theta\|^2_{H_p - \bar{a}_p \bar{a}_p^\top} = \max_{\tilde{a}^\star \in \tilde{\mathcal{A}}} \sup_{\tilde{\theta} \in \mathbb{R}^{\dim(\mathcal{A})}} \langle \bar{a}_{\tilde{p}} - \tilde{a}^\star, \tilde{\theta} \rangle - \frac{\gamma}{4} \|P(\hat{\theta}) - \tilde{\theta}\|^2_{H_{\tilde{p}} - \bar{a}_{\tilde{p}} \bar{a}_{\tilde{p}}^\top}.$$

**C.2.2   Handling the case where** $\dim(\mathcal{A}) = d$**.**

**Lemma 4.** When $\dim(\mathcal{A}) = d$, any solution $p \in \Delta(\mathcal{A})$ to the problem logdet-barrier$(\theta, \gamma; \mathcal{A})$ in Eq. (6) satisfies

$$\forall a \in \mathcal{A}: \; \langle \bar{a}_p - a, \theta \rangle + \frac{1}{\gamma} \|\bar{a}_p - a\|^2_{H_p^{-1} - \bar{a}_p \bar{a}_p^\top} \le \frac{\dim(\mathcal{A})}{\gamma}.$$

**Proof.** We first observe that any solution $p \in \Delta(\mathcal{A})$ to the problem logdet-barrier$(\hat{\theta}, \gamma; \mathcal{A})$ must be positive definite in the sense that $H_p - \bar{a}_p \bar{a}_p^\top \succ 0$, since otherwise the objective has value $\infty$; note that $\dim(\mathcal{A}) = d$ implies that there exists $p$ with $H_p - \bar{a}_p \bar{a}_p^\top \succ 0$. Going forward we work only with $p$ for which $H_p - \bar{a}_p \bar{a}_p^\top \succ 0$.

Recall $p = $ logdet-barrier$(\hat{\theta}, \gamma; \mathcal{A})$ is any solution to

$$\underset{p \in \Delta(\mathcal{A})}{\operatorname{argmin}} \left\{ \langle \bar{a}_p, \hat{\theta} \rangle - \gamma^{-1} \log \det(H_p - \bar{a}_p \bar{a}_p^\top) \right\},$$

where $\Delta(\mathcal{A})$ is the set of distributions over countable subsets of $\mathcal{A}$. Hence we can write

$$\Delta(\mathcal{A}) = \left\{ \sum_{i=1}^\infty w_i \mathbf{e}_{A_i} \mid w \in \mathbb{R}_+^{\mathbb{N}}, A \in \mathcal{A}^{\mathbb{N}}, \sum_{i=1}^\infty w_i = 1 \right\},$$

where $\mathbf{e}_a$ denotes the distribution that selects $a$ with probability 1. By first-order optimality, $p$ is a solution to Eq. (6) if and only if

$$\forall p' \in \Delta(\mathcal{A}): \sum_{a \in \operatorname{supp}(p) \cup \operatorname{supp}(p')} (p'_a - p_a) \frac{\partial}{\partial p_a} \left[ \langle \bar{a}_p, \hat{\theta} \rangle - \frac{1}{\gamma} \log \det(H_p - \bar{a}_p \bar{a}_p^\top) \right] \ge 0.$$

By the K.K.T. conditions, this is the case if and only if there exists some $\tilde{\lambda} \in \mathbb{R}$ such that

$$\forall a \in \operatorname{supp}(p): \frac{\partial}{\partial p_a} \left[ \langle \bar{a}_p, \hat{\theta} \rangle - \frac{1}{\gamma} \log \det(H_p - \bar{a}_p \bar{a}_p^\top) \right] = \tilde{\lambda} \tag{14}$$

$$\forall a \in \mathcal{A}: \frac{\partial}{\partial p_a} \left[ \langle \bar{a}_p, \hat{\theta} \rangle - \frac{1}{\gamma} \log \det(H_p - \bar{a}_p \bar{a}_p^\top) \right] \ge \tilde{\lambda}. \tag{15}$$

To find $\tilde{\lambda}$, we calculate the partial derivative with the chain rule:

$$\frac{\partial}{\partial p_a} \left[ \langle \bar{a}_p, \hat{\theta} \rangle - \frac{1}{\gamma} \log \det(H_p - \bar{a}_p \bar{a}_p^\top) \right]$$

$$= \langle a, \hat{\theta} \rangle - \frac{\det(H_p - \bar{a}_p \bar{a}_p^\top) \operatorname{tr}((H_p - \bar{a}_p \bar{a}_p^\top)^{-1} (aa^\top - \bar{a}_p a^\top - a\bar{a}_p^\top))}{\gamma \det(H_p - \bar{a}_p \bar{a}_p^\top)}$$

$$= \langle a - \bar{a}_p, \hat{\theta} \rangle - \frac{1}{\gamma} \|a - \bar{a}_p\|^2_{(H_p - \bar{a}_p \bar{a}_p^\top)^{-1}} + \frac{1}{\gamma} \|\bar{a}_p\|^2_{(H_p - \bar{a}_p \bar{a}_p^\top)^{-1}} + \langle \bar{a}_p, \hat{\theta} \rangle.$$

Using Eq. (14) and taking the expectation over $p$ yields

$$\tilde{\lambda} = \mathbb{E}_{a \sim p} \left[ \frac{\partial}{\partial p_a} \left[ \langle \bar{a}_p, \hat{\theta} \rangle - \frac{1}{\gamma} \log \det(H_p - \bar{a}_p \bar{a}_p^\top) \right] \right] = -\frac{d}{\gamma} + \frac{1}{\gamma} \|\bar{a}_p\|^2_{(H_p - \bar{a}_p \bar{a}_p^\top)^{-1}} + \langle \bar{a}_p, \hat{\theta} \rangle.$$

Finally, plugging this into Eq. (15), we get

$$\forall a \in \mathcal{A}: \langle a - \bar{a}_p, \hat{\theta} \rangle - \frac{1}{\gamma} \|a - \bar{a}_p\|^2_{(H_p - \bar{a}_p \bar{a}_p^\top)^{-1}} \ge -\frac{d}{\gamma}.$$

Rearranging finishes the proof. $\qquad\square$

**Proof of Lemma 3.** As mentioned in the previous proof, for any solution $p \in \Delta(\mathcal{A})$ to the problem logdet-barrier$(\hat{\theta}, \gamma; \mathcal{A})$ the matrix $H_p - \bar{a}_p \bar{a}_p^\top$ is positive definite. In this case, for any fixed $a^\star \in \mathcal{A}$, the function

$$\theta \mapsto \langle \bar{a}_p - a^\star, \theta \rangle - \frac{\gamma}{4} \|\hat{\theta} - \theta\|^2_{H_p - \bar{a}_p \bar{a}_p^\top}$$

is strictly concave in $\theta$ and the maximizer $\theta^\star$ is found by setting the derivative with respect to $\theta$ to 0:

$$\frac{\partial}{\partial\theta}\left[\langle\bar{a}_p - a^\star, \theta\rangle - \frac{\gamma}{4}\|\hat{\theta} - \theta\|^2_{H_p - \bar{a}_p\bar{a}_p^\top}\right] = \bar{a}_p - a^\star + \frac{\gamma}{2}(H_p - \bar{a}_p\bar{a}_p^T)(\hat{\theta} - \theta)$$

$$\theta^\star = \hat{\theta} + \frac{2}{\gamma}(H_p - \bar{a}_p\bar{a}_p^\top)^{-1}(a_p - a^\star).$$

Substituting in this choice, we have that

$$\max_{a^\star\in\mathcal{A}}\sup_{\theta\in\mathbb{R}^d}\langle\bar{a}_p - a^\star, \theta\rangle - \frac{\gamma}{4}\|\hat{\theta} - \theta\|^2_{H_p - \bar{a}_p\bar{a}_p^\top} = \max_{a^\star\in\mathcal{A}}\langle\bar{a}_p - a^\star, \hat{\theta}\rangle + \frac{1}{\gamma}\|\bar{a}_p - a\|^2_{(H_p - \bar{a}_p\bar{a}_p^\top)^{-1}}.$$

To complete the proof, we apply Lemma 4 to the right-hand side above. $\qquad\square$

### C.3 Proofs from Section 3.3

**Proof of Theorem 3.** Let $m$ be fixed. To keep notation compact, we abbreviate $q_t \equiv q_{t,h}$, $\rho_t \equiv \rho_{t,m}$, $\gamma_t \equiv \gamma_{t,m}$, $Z_t \equiv Z_{t,m}$, and so forth.

Let the sequence $S$ be fixed, and let $f^\star$ be any predictor achieving the value of $\varepsilon_T(S)$. If the infimum is not achieved, we can consider a limit sequence; we omit the details. Recall that since we assume an oblivious adversary, $f^\star$ is fully determined before the interaction protocol begins. Let us abbreviate $\theta_t^\star = f^\star(x_t)$, $a_t^\star = \pi_{f^\star}(x_t)$, and $\pi_t^\star(x_t) = \mathrm{argmin}_{a\in\mathcal{A}_t}\mu(a, x_t)$, where ties are broken arbitrarily. Then we can bound

$$\mathrm{Reg}_{\mathrm{Imp}}(T) = \mathbb{E}\left[\sum_{t=1}^T \frac{Z_t}{q_t}\left(\mu(a_t, x_t) - \mu(\pi_t^\star(x_t), x_t)\right)\right]$$

$$\leq \mathbb{E}\left[\sum_{t=1}^T \frac{Z_t}{q_t}\left(\langle a_t - \pi_t^\star(x_t), \theta_t^\star\rangle + 2\max_{a\in\mathcal{A}_t}|\mu(a, x_t) - \langle a, \theta_t^\star\rangle|\right)\right]$$

$$\overset{(a)}{\leq} \mathbb{E}\left[\sum_{t=1}^T \frac{Z_t}{q_t}\left(\langle a_t - \pi_t^\star(x_t), \theta_t^\star\rangle\right)\right] + 2\varepsilon_T T$$

$$\overset{(b)}{\leq} \mathbb{E}\left[\sum_{t=1}^T \frac{Z_t}{q_t}\langle a_t - a_t^\star, \theta_t^\star\rangle\right] + 2\varepsilon_T T$$

$$\overset{(c)}{=} \mathbb{E}\left[\sum_{t=1}^T \frac{Z_t}{q_t}\left(\langle\bar{a}_{p_t} - a_t^\star, \theta_t^\star\rangle - \frac{\gamma_t}{4}\|\hat{\theta}_t - \theta^\star\|^2_{H_{p_t}} + \frac{\gamma_t}{4}\|\hat{\theta}_t - \theta^\star\|^2_{H_{p_t}}\right)\right] + 2\varepsilon_T T$$

$$\overset{(d)}{\leq} \mathbb{E}\left[\sum_{t=1}^T \frac{Z_t}{q_t}\left(\frac{\mathrm{dim}(\mathcal{A}_t)}{\gamma_t} + \frac{\gamma_t}{4}\|\hat{\theta}_t - \theta^\star\|^2_{H_{p_t}}\right)\right] + 2\varepsilon_T T$$

$$\overset{(e)}{\leq} \mathbb{E}\left[\max_{t\in[T]}\gamma_t^{-1}\right]\sum_{t=1}^T \mathrm{dim}(\mathcal{A}_t) + \mathbb{E}\left[\sum_{t=1}^T \frac{Z_t}{q_t}\frac{\gamma_t}{4}(\langle a_t, \hat{\theta}_t\rangle - \langle a_t, \theta_t^\star\rangle)^2\right] + 2\varepsilon_T T.$$

Here (a) follows from the fact that $\mathbb{E}[Z_t] = q_t$ and the Cauchy-Schwarz inequality, together with the definition of $\varepsilon_T$; (b) follows from the definition of the policy $\pi_{f^\star}$; (c) is due to the fact that, conditioned on $Z_t = 1$, we sample $a_t \sim p_t$ with $\mathbb{E}_{a_t\sim p_t}[a_t] = \bar{a}_{p_t}$; (d) uses Lemma 2; (e) uses $\mathbb{E}_{a_t\sim p_t}[a_t a_t^\top] = H_{p_t}$. Continuing with squared error term above, we have

$$\mathbb{E}\left[\sum_{t=1}^T \frac{Z_t}{q_t}\gamma_t(\langle a_t, \hat{\theta}_t\rangle - \langle a_t, \theta_t^\star\rangle)^2\right]$$

$$= \mathbb{E}\left[\sum_{t=1}^T \frac{Z_t}{q_t}\gamma_t\left((\langle a_t, \hat{\theta}_t\rangle - \ell_t)^2 - (\langle a_t, \theta_t^\star\rangle - \ell_t)^2 + 2(\ell_t - \langle a_t, \theta_t^\star\rangle)\langle a_t, \hat{\theta}_t - \theta_t^\star\rangle\right)\right]$$

$$\overset{(a)}{=} \mathbb{E}\left[\sum_{t=1}^T \frac{Z_t}{q_t}\gamma_t\left((\langle a_t, \hat{\theta}_t\rangle - \ell_t)^2 - (\langle a_t, \theta_t^\star\rangle - \ell_t)^2 + 2(\mu(a_t, x_t) - \langle a_t, \theta_t^\star\rangle)\langle a_t, \hat{\theta}_t - \theta_t^\star\rangle\right)\right],$$

where (a) uses that $\ell_t$ is conditionally independent of $Z_t$ and $a_t$. We bound the term involving the difference of squares as

$$\mathbb{E}\left[\sum_{t=1}^{T}\frac{Z_t}{q_t}\gamma_t((\langle a_t,\hat{\theta}_t\rangle-\ell_t)^2-(\langle a_t,\theta_t^{\star}\rangle-\ell_t)^2)\right]\leq\mathbb{E}\left[\max_{t\in[T]}\frac{\gamma_t}{q_t}\right]\mathsf{Reg}_{\mathrm{Sq}}(T),$$

by Assumption 2, which also holds if SqAlg runs for less than $T$ timesteps, since we could extend the sequence with 0 weight until time $T$. For the linear term, we apply the sequence of inequalities

$$2\mathbb{E}\left[\sum_{t=1}^{T}\frac{Z_t}{q_t}\gamma_t(\mu(a_t,x_t)-\langle a_t,\theta_t^{\star}\rangle)\langle a_t,\hat{\theta}_t-\theta_t^{\star}\rangle\right]$$

$$\stackrel{(a)}{\leq}2\mathbb{E}\left[\sum_{t=1}^{T}\frac{Z_t}{q_t}\gamma_t((\mu(a_t,x_t)-\langle a_t,\theta_t^{\star}\rangle)^2+\frac{1}{4}\langle a_t,\hat{y}_t-\theta_t^{\star}\rangle^2\right]$$

$$\leq2\mathbb{E}\left[\sum_{t=1}^{T}\frac{Z_t}{q_t}\gamma_t\max_{a\in\mathcal{A}_t}((\mu(a,x_t)-\langle a,\theta_t^{\star}\rangle)^2+\frac{1}{4}\langle a_t,\hat{y}_t-\theta_t^{\star}\rangle^2\right]$$

$$\stackrel{(b)}{\leq}2\mathbb{E}\left[\max_{t\in[T]}\gamma_t\right]\varepsilon_T^2 T+\frac{1}{2}\mathbb{E}\left[\sum_{t=1}^{T}\frac{Z_t}{q_t}\gamma_t(\langle a_t,\hat{\theta}_t\rangle-\langle a_t,\theta_t^{\star}\rangle)^2\right],$$

where (a) is by the AM-GM inequality: $2ab\leq 2a^2+\frac{1}{2}b^2$; (b) follows from the fact that $Z_t$ is conditionally independent of $\gamma_t$, and the definition of $\varepsilon_T$.

Altogether, we have

$$\mathbb{E}\left[\sum_{t=1}^{T}\frac{Z_t}{q_t}\gamma_t(\langle a_t,\hat{\theta}_t\rangle-\langle a_t,\theta_t^{\star}\rangle)^2\right]$$

$$\leq\mathbb{E}\left[\max_{t\in[T]}\frac{\gamma_t}{q_t}\right]\mathsf{Reg}_{\mathrm{Sq}}(T)+2\mathbb{E}\left[\max_{t\in[T]}\gamma_t\right]\varepsilon_T^2 T+\frac{1}{2}\mathbb{E}\left[\sum_{t=1}^{T}\frac{Z_t}{q_t}\gamma_t(\langle a_t,\hat{\theta}_t\rangle-\langle a_t,\theta_t^{\star}\rangle)^2\right].$$

Rearranging yields

$$\mathbb{E}\left[\sum_{t=1}^{T}\frac{Z_t}{q_t}\gamma_t(\langle a_t,\hat{\theta}_t\rangle-\langle a_t,\theta_t^{\star}\rangle)^2\right]\leq2\mathbb{E}\left[\max_{t\in[T]}\frac{\gamma_t}{q_t}\right]\mathsf{Reg}_{\mathrm{Sq}}(T)+4\mathbb{E}\left[\max_{t\in[T]}\gamma_t\right]\varepsilon_T^2 T.$$

Combining all of the developments so far, we have

$$\mathsf{Reg}_{\mathrm{Imp}}(T)\leq\sum_{t=1}^{T}\mathbb{E}\left[\gamma_t^{-1}\right]\dim(\mathcal{A}_t)+\frac{1}{2}\mathbb{E}\left[\max_{t\in[T]}\frac{\gamma_t}{q_t}\right]\mathsf{Reg}_{\mathrm{Sq}}(T)+\mathbb{E}\left[\max_{t\in[T]}\gamma_t\right]\varepsilon_T^2 T+2\varepsilon_T T.$$
(16)

The proof is completed by noting that the learning rate $\gamma_t=\min\left\{\frac{\sqrt{d}}{\varepsilon'},\sqrt{dT/(\rho_t\mathsf{Reg}_{\mathrm{Sq}}(T))}\right\}$ is non-increasing, but $\gamma_t\rho_t\geq\frac{\gamma_t}{q_t}$ is non-decreasing. Hence, we can upper bound the expression above by

$$\mathsf{Reg}_{\mathrm{Imp}}(T)\leq\mathbb{E}\left[\gamma_T^{-1}\right]dT+\frac{1}{2}\mathbb{E}\left[\gamma_T\rho_T\right]\mathsf{Reg}_{\mathrm{Sq}}(T)+\mathbb{E}[\gamma_1]\varepsilon_T^2 T+2\varepsilon_T T$$

$$\leq\left(\frac{\varepsilon'}{\sqrt{d}}+\mathbb{E}[\sqrt{\rho_T}]\sqrt{\frac{\mathsf{Reg}_{\mathrm{Sq}}(T)}{dT}}\right)dT+\frac{1}{2}\mathbb{E}[\sqrt{\rho_T}]\sqrt{dT\mathsf{Reg}_{\mathrm{Sq}}(T)}+\frac{\sqrt{d}}{\varepsilon'}\varepsilon_T^2 T+2\varepsilon_T T.$$

$\square$

**Proof of Theorem 1.** Let $m^{\star}:=\operatorname{argmin}_{m\in[M]}\frac{\varepsilon_T}{\varepsilon_m'}+\frac{\varepsilon_m'}{\varepsilon_T}$ if $\varepsilon_T\geq T^{-1}$ and $m^{\star}=M$ otherwise. We begin by formally verifying the claim

$$\mathsf{Reg}(T)=\mathbb{E}\left[\sum_{t=1}^{T}\tilde{\ell}_{t,A_t}-\tilde{\ell}_{t,m^{\star}}\right]+\mathsf{Reg}_{\mathrm{Imp}}^{m^{\star}}(T).$$
(17)

By the definition $\tilde{\ell}_{t,A_t} := \ell_t + 1$, we have

$$\mathbb{E}\left[\tilde{\ell}_{t,A_t} - \tilde{\ell}_{t,m^\star}\right] = \mathbb{E}\left[\ell_t + 1 - \frac{Z_{t,m^\star}}{p_{t,m^\star}}(\ell_t + 1)\right] = \mathbb{E}\left[\mu(a_t, x_t) - \frac{Z_{t,m^\star}}{p_{t,m^\star}}\mu(a_t, x_t)\right].$$

The second term is

$$\mathsf{Reg}_{\mathrm{Imp}}^{m^\star}(T) = \mathbb{E}\left[\sum_{t=1}^{T} \frac{Z_{t,m^\star}}{p_{t,m^\star}}(\mu(a_t, x_t) - \mu(\pi_t^\star(x_t), x_t))\right] = \mathbb{E}\left[\sum_{t=1}^{T} \frac{Z_{t,m^\star}}{p_{t,m^\star}}\mu(a_t, x_t) - \mu(\pi_t^\star(x_t), x_t)\right].$$

Combining both lines leads to Eq. (17). The losses $\tilde{\ell}$ satisfy $\forall m \in [M] : \tilde{\ell}_{t,m} \in [0,2]$, since $\ell_t \in [-1,1]$ and we shift the loss by 1. Hence we can apply Corollary 2 with $\alpha = \frac{1}{2}$ and $R = \frac{3}{2}\sqrt{dT\mathsf{Reg}_{\mathrm{Sq}}(T)}$ to obtain

$$\mathbb{E}\left[\sum_{t=1}^{T} \tilde{\ell}_{t,A_t} - \tilde{\ell}_{t,m^\star}\right] \leq 4\sqrt{2MT} + 3\sqrt{dT\mathsf{Reg}_{\mathrm{Sq}}(T)M} - \frac{3}{2}\mathbb{E}[\sqrt{\rho_{T,a^\star}}]\sqrt{dT\mathsf{Reg}_{\mathrm{Sq}}(T)},$$

and by Theorem 3,

$$\mathsf{Reg}_{\mathrm{Imp}}^{m^\star}(T) \leq \left(\left(\frac{\varepsilon'_{m^\star}}{\varepsilon_T} + \frac{\varepsilon_T}{\varepsilon'_{m^\star}}\right)\sqrt{d} + 2\right)\varepsilon_T T + \frac{3}{2}\mathbb{E}[\sqrt{\rho_{T,a^\star}}]\sqrt{dT\mathsf{Reg}_{\mathrm{Sq}}(T)}.$$

Either $\varepsilon_T > T^{-1}$, in which case we can pick $m^\star$ such that $\varepsilon'_{m^\star} \in [\varepsilon_T, e\varepsilon_T]$ and $\left(\frac{\varepsilon'_{m^\star}}{\varepsilon_T} + \frac{\varepsilon_T}{\varepsilon'_{m^\star}}\right) \leq e + e^{-1}$, or we pick $\varepsilon'_{m^\star} = T^{-1}$ and the misspecification term is bounded by

$$\left(\left(\frac{\varepsilon'_{m^\star}}{\varepsilon_T} + \frac{\varepsilon_T}{\varepsilon'_{m^\star}}\right)\sqrt{d} + 2\right)\varepsilon_T T = \left(\left(\varepsilon'_{m^\star} + \frac{\varepsilon_T^2}{\varepsilon'_{m^\star}}\right)\sqrt{d} + 2\varepsilon_T\right)T \leq 2\sqrt{d} + 2.$$

Summing the regret bounds for the base and master algorithms completes the proof. $\qquad\square$

## C.4  Proofs from Section 3.6

The procedure runs in episodes. At the begin of episode 1, the algorithm assumes $D_1 = \sum_{t=1}^{T} \dim(\mathcal{A}_t) \leq 2T$ and initializes its learning rate accordingly. Within each episode $i \geq 1$, if the agent observes at time $t$ that $\sum_{s=\tau_i}^{t} \dim(\mathcal{A}_s) > D_i$, it restarts the algorithm with $D_{i+1} = 2D_i$; we denote this time by $\tau_{i+1} = t$. We can assume $\dim(\mathcal{A}) \leq d < T$ or the result is trivial, hence we never need to double more than once at each time step. For technical reasons, we require that the bound of Theorem 1 also holds when the agent plays only on a subset of time steps.

**Corollary 3.** *Let $\mathcal{T} \subset [T]$ be a subset of the time horizon chosen oblivious to the actions of the agent. Then the upper bound of Theorem 1 for* SquareCB.Imp *on the sequence $S$ is also an upper bound for running the algorithm on the sub-sequence $S_{\mathcal{T}}$. This also holds for any refinement of the bound based on the average dimension $d_{\mathrm{avg}}$ instead of $d$.*

**Proof.** We extend the sequence $S$ by adding an "end" sequence $E = (\{0\}, \mathfrak{x})_{t=1}^{T}$, where $\mathfrak{x} \in \mathcal{X}$ is picked such that $\mu(0, \mathfrak{x}) = 0$ (If there is no such context, we add a context with that property to $\mathcal{X}$). Let the enhanced sequence be $S' = S + E$ and consider the sequence $\tilde{S} = S'_{\mathcal{T} \cup \{T+1,\ldots,2T-|\mathcal{T}|\}}$, which is of length $T$. The regret contribution from playing on the $E$ section on the sequence is always 0, since there is only one action. Furthermore $\varepsilon_T(\tilde{S}) \leq \varepsilon_T(S)$ Hence we have by Theorem 1

$$\mathbb{E}\left[\sum_{t\in\mathcal{T}} \mu(a_t, x_t) - \min_{a\in\mathcal{A}_t} \mu(a, x_t)\right] = \mathbb{E}\left[\sum_{t\in\mathcal{T}\{T+1,\ldots,2T-|\mathcal{T}|\}} \mu(a_t, x_t) - \min_{a\in\mathcal{A}_t} \mu(a, x_t)\right]$$
$$\leq \mathcal{O}\left(\sqrt{d}\varepsilon_T(S)T + \sqrt{d\mathsf{Reg}_{\mathrm{Sq}}(T)\log(T)}\right).$$

Since $d_{\mathrm{avg}}(\tilde{S}) \leq d_{\mathrm{avg}}(S)$, this argument also extends to the refined version where $d$ is replaced by $d_{\mathrm{avg}}$ if the algorithm parameters are tuned accordingly. $\qquad\square$

**Proof of Theorem 4.** Let $\tau_1, \ldots, \tau_L$ denote the times where the algorithm is restarted, with $\tau_1 = 1$ and $\tau_{L+1} = T + 1$ by convention.

Since the adversary fixes the action sets in advance, these doubling times are deterministic. The regret is given by

$$\mathsf{Reg}(T) = \mathbb{E}\left[\sum_{t=1}^{T} \mu(a_t, x_t) - \min_{a \in \mathcal{A}_t} \mu(a, x_t)\right]$$

$$\leq \sum_{i=1}^{L} \mathbb{E}\left[\sum_{t=\tau_i}^{\tau_{i+1}-1} \mu(a_t, x_t) - \min_{a \in \mathcal{A}_t} \mu(a, x_t)\right].$$

By Corollary 3 applied to each episode

$$\mathbb{E}\left[\sum_{t=\tau_{i-1}+1}^{\tau_i} \mu(a_t, x_t) - \min_{a \in \mathcal{A}_t} \mu(a, x_t)\right] = \sqrt{2^i} \cdot \mathcal{O}\left(\varepsilon_T(S)T + \sqrt{\mathsf{Reg}_{\mathsf{Sq}}(T)\log(T)}\right).$$

Summing over these terms and observing that

$$\sum_{i=1}^{L} 2^{i/2} = \mathcal{O}(2^{L/2}) = \mathcal{O}(1) \cdot \sqrt{\frac{1}{T}\sum_{t=1}^{T}\dim(\mathcal{A}_t)} = \mathcal{O}\left(d_{\mathrm{avg}}^{1/2}\right),$$

completes the proof. $\qquad\square$

## D  Improved Master Algorithms for Bandit Aggregation

In this section, we present a general class of algorithms that can be used for the master algorithm within the framework of Algorithm 3. For the remainder of this section, we are working in a generic adversarial multi-armed bandit setting, where the agent selects an action $A_t \in [M]$ at any time step and observes the associated loss $\ell_{t,A_t} \in [0, L]$. Compared to the original CORRAL algorithm of Agarwal et al. [9], our new algorithms are simpler to analyze, more flexible, and improve in terms of logarithmic factors.

The CORRAL algorithm is a special case of Algorithm 3 that uses a bandit variant of Online Mirror Descent (OMD) algorithm with log-barrier regularization as the master.[6]  The bandit variant of the OMD algorithm used within CORRAL is parameterized by a Legendre potential $F(x) = \sum_{i=1}^{d} \eta_i^{-1} f(x_i)$ where $\eta_1, \ldots, \eta_d$ are per-coordinate learning rates. It initializes the distribution $p_1 = \mathrm{argmin}_{p \in \Delta([M])} F(p)$. At each time $t$, the bandit OMD algorithm samples arm $A_t \sim p_t$, observes $\ell_t$, and constructs an unbiased importance-weighted loss estimator $\hat{\ell}_t = \frac{\ell_{t,A_t}}{p_{t,A_t}} \mathbf{e}_{A_t}$. It then updates the action distribution as

$$p_{t+1} = \mathrm{argmin}_{p \in \Delta([M])} \langle p, \hat{\ell} \rangle + D_F(p, p_t), \tag{18}$$

where $D_F(x, y) := F(x) - F(y) - \langle x - y, \nabla F(y) \rangle$ is the Bregman divergence associated with $F$.

A key to the performance of the CORRAL master is an time-dependent learning rate[7] schedule for each of the per-arm learning rates, which increases the learning rate for each arm whenever the probability for that arm falls below a certain threshold.

An algorithm closely related to OMD is the Follow-the-Regularized-Leader (FTRL) algorithm. In particular, for any sequence of loss vector estimates $(\hat{\ell}_t)_{t=1}^{T}$, there exists a sequence of (vector) biases $b_t$, such that FTRL running on the loss sequence $(\hat{\ell}_t - b_t)_{t=1}^{T}$ using the same learning rate as its OMD counterpart has an identical trajectory of plays $p_t$.

We can view the CORRAL master through the lens of FTRL: the algorithm performs two steps whenever it increases the learning rate of arm $i$. First it subtracts a bias $b_{t,i} > 0$ from the loss

estimates for arm $i$. Then it increases the learning rate for that arm. We argue that only the former step is actually relevant to the performance of CORRAL, while the latter is unnecessary, and ends up complicating the analysis. This motivates the $(\alpha, R)$-hedged FTRL algorithm, which achieves a slightly improved guarantee by removing the per-coordinate learning rates.

## D.1 The Hedged FTRL Algorithm

Following the intuition in the prequel we propose $(\alpha, R)$-hedged FTRL, a modified variant of the FTRL algorithm with strong guarantees for aggregating bandit algorithms. We begin by defining a basic bandit variant of FTRL algorithm.

The FTRL family algorithms of algorithms is parameterized by a potential $F$ and learning rate $\eta > 0$. At each round $t$, the algorithm selects

$$p_t = \mathrm{argmin}_{p \in \Delta([M])} \langle p, \hat{L}_{t-1} \rangle + \eta^{-1} F(p), \quad \text{where} \quad \hat{L}_t = \sum_{s=1}^{t} \hat{\ell}_s.$$

Two relevant properties of $F$ that arise in our analysis are *stability* and *diameter*. Define

$$\bar{F}_\eta^\star(-L) = \max_{p \in \Delta([M])} \langle p, -L \rangle - \eta^{-1} F(p).$$

The stability $\mathrm{stab}(F)$ and diameter $\mathrm{diam}(F)$ of $F$ for a loss range $[0, L]$ are define as follows:

$$\mathrm{stab}(F) = \sup_{\eta > 0} \sup_{x \in \Delta([M])} \sup_{\ell \in [0,L]^M} \eta^{-1} \mathbb{E}_{A \sim x} \left[ D_{\bar{F}_\eta^\star} \left( \eta^{-1} \nabla F(x) - \frac{\ell_A}{x_A} \mathbf{e}_A, \eta^{-1} \nabla F(x) \right) \right],$$

$$\mathrm{diam}(F) = \max_{p \in \Delta([M])} F(p) - \min_{p \in \Delta([M])} F(p).$$

Given a potential with bounded $\mathrm{stab}(F)$ and $\mathrm{diam}(F)$, setting the learning rate as $\eta = \sqrt{\mathrm{diam}(F)/(\mathrm{stab}(F)T)}$ leads to a regret bound $2\sqrt{\mathrm{stab}(F) \mathrm{diam}(F) T}$ for FTRL [5].[8] Well-known algorithms that arise as special cases of this result include:

- EXP3 [13] is an instance of FTRL with $F(x) = \sum_{i=1}^{M} x_i \log(x_i)$, $\mathrm{diam}(F) = \log(M)$ and $\mathrm{stab}(F) \leq \frac{L^2 M}{2}$.
- Tsallis-INF [11, 5, 48] is the instance of FTRL with the best known regret bound. It is given by $F(x) = -2 \sum_{i=1}^{M} \sqrt{x_i}$ with $\mathrm{diam}(F) \leq 2\sqrt{M}$ and $\mathrm{stab}(F) \leq L^2 \sqrt{M}$.

We can now present the $(\alpha, R)$-hedged FTRL algorithm. The algorithm augments the basic FTRL strategy using an additional pair of parameters $(\alpha, R) \in (0, 1) \times \mathbb{R}$. The algorithm initializes $(B_{0,i})_{i=1}^{M}$ with $B_{0,i} = \rho_{1,i}^{\alpha} R$. At each step $t$, it plays $A_t \sim p_t$ and computes

$$\tilde{p}_{t+1} = \mathrm{argmin}_{p \in \Delta([M])} \langle p, \hat{L}_t - (B_{t-1} - B_0) \rangle + \eta^{-1} F(p), \quad \text{where} \quad \hat{L}_t = \sum_{s=1}^{t} \hat{\ell}_s.$$

If $\tilde{p}_{t+1,A_t}^{-\alpha} R \leq B_{t-1,A_t}$, the algorithm sets $B_t = B_{t-1}$ and $p_{t+1} = \tilde{p}_{t+1}$. Otherwise it chooses the unique $b_t > 0$, such that for $B_t = B_{t-1} + b_t \mathbf{e}_{A_t}$ it holds simultaneously

$$p_{t+1} = \mathrm{argmin}_{p \in \Delta([M])} \langle p, \hat{L}_t - (B_t - B_0) \rangle + \eta^{-1} F(p) \qquad \text{and} \qquad p_{t+1,A_t}^{-\alpha} R = B_{t,A_t}.$$

This algorithm is always well defined (see Appendix D.2 for details). The main regret guarantee is as follows. Let $\rho_{t,i} = \max_{s \leq t} p_{s,i}^{-1}$.

**Theorem 6.** *Then for any potential $F$ with $\mathrm{stab}(F), \mathrm{diam}(F) < \infty$, the pseudo-regret $\mathsf{Reg}_M(T) = \mathbb{E}\left[ \sum_{t=1}^{T} \ell_{t,A_t} - \ell_{t,a^\star} \right]$ of $(\alpha, R)$-hedged FTRL run with learning rate $\eta = \sqrt{\mathrm{diam}(F)/(\mathrm{stab}(F)T)}$ against any arm $a^\star \in [M]$ is bounded as follows:*

$$\mathsf{Reg}_M(T) \leq 2\sqrt{\mathrm{stab}(F) \mathrm{diam}(F) T} + \left[ \frac{\alpha}{1-\alpha} \sum_{i=1}^{M} \left( \rho_{1,i}^{\alpha-1} - \mathbb{E}[\rho_{T,i}^{\alpha-1}] \right) + \rho_{1,a^\star}^{\alpha} - \mathbb{E}[\rho_{T,a^\star}^{\alpha}] \right] \cdot R.$$

The algorithm may be viewed "hedging" against the event that the arm $a^\star$ experiences a very small probability, as this guarantees a negative regret contribution of $\rho_{T,a^\star}^{-\alpha} R$.

## D.2 Proofs

Before proving the main result, we first established that the $(\alpha, R)$-hedged FTRL is well-defined. The algorithm initializes with $B_0$ such that $\nabla \bar{F}_\eta^\star(B_0)_i^{-\alpha} R = B_{0,i}$. For symmetric potentials $F(x) = \sum_{i=1}^M f(x_i)$, $\nabla \bar{F}_\eta^\star(c\mathbf{1}_M) = \frac{1}{M}\mathbf{1}_M$ for any $c \in \mathbb{R}$. Hence $B_0 = M^{-\alpha} R \mathbf{1}_M$ satisfies the initialization condition. Otherwise a solution exists by the observation that $\nabla \bar{F}_\eta^\star(B_0)_i^{-\alpha} R$ is a continuous, decreasing function in $B_{0,i}$ that has positive values at $B_0 = 0$. Hence a solution to the equation must exist.

The same argument holds during the update at subsequent rounds $t$. Only the arm that was played can decrease in probability, which means we only need to ensure that $\rho_{t+1,A_t}^\alpha R = B_{t,A_t}$. The LHS is continuously decreasing with increasing $b_t$, while the RHS is increasing. The optimal value must exist, it is unique and lays in $[0, \hat{\ell}_{t,A_t}]$.

**Proof of Theorem 6.** We follow the standard FTRL analysis. Let $\tilde{B}_t = B_t - B_0$ and note that $p_t = \nabla \bar{F}_\eta^\star(-\hat{L}_{t-1} + \tilde{B}_{t-1})$, so $\langle p_t, \hat{\ell}_t \rangle = \langle \nabla \bar{F}_\eta^\star(-\hat{L}_{t-1} + \tilde{B}_{t-1}), \hat{L}_t - \hat{L}_{t-1} \rangle$. Hence, we can write

$$
\mathbb{E}\left[\sum_{t=1}^T \ell_{t,A_t} - \ell_{t,a^\star}\right] = \mathbb{E}\left[\sum_{t=1}^T \langle p_t, \hat{\ell}_t \rangle - \hat{\ell}_{t,a^\star}\right]
$$

$$
= \mathbb{E}\left[\sum_{t=1}^T D_{\bar{F}_\eta^\star}(-\hat{L}_t + \tilde{B}_{t-1}, -\hat{L}_{t-1} + \tilde{B}_{t-1})\right]
$$

$$
+ \mathbb{E}\left[\sum_{t=1}^T \left(-\bar{F}_\eta^\star(-\hat{L}_t + \tilde{B}_{t-1}) + \bar{F}_\eta^\star(-\hat{L}_{t-1} + \tilde{B}_{t-1}) - \hat{\ell}_{t,a^\star}\right)\right].
$$

Note that there exists $\lambda$ such that $-\hat{L}_{t-1} + \tilde{B}_{t-1} = \lambda \mathbf{1}_M + \eta^{-1} \nabla F(p_t)$. Furthermore, adding or subtracting the same $\lambda \mathbf{1}_M$ term to both arguments does not change the value of the Bregman divergence, because $\bar{F}_\eta(-L + \lambda \mathbf{1}_M) = F_\eta(-L) + \lambda$. Thus,

$$
\mathbb{E}\left[\sum_{t=1}^T D_{\bar{F}_\eta^\star}(-\hat{L}_t + \tilde{B}_{t-1}, -\hat{L}_{t-1} + \tilde{B}_{t-1})\right]
$$

$$
= \eta \mathbb{E}\left[\sum_{t=1}^T \eta^{-1} D_{\bar{F}_\eta^\star}(\eta^{-1} \nabla F(p_t) - \hat{\ell}_t, \eta^{-1} \nabla F(p_t))\right] \leq \eta \operatorname{stab}(F) T.
$$

Rearranging the second term gives

$$
\sum_{t=1}^T \left(-\bar{F}_\eta^\star(-\hat{L}_t + \tilde{B}_{t-1}) + \bar{F}_\eta^\star(-\hat{L}_{t-1} + \tilde{B}_{t-1}) - \hat{\ell}_{t,a^\star}\right)
$$

$$
= \bar{F}_\eta^\star(0) - \bar{F}_\eta^\star(-\hat{L}_T + \tilde{B}_{T-1}) - \hat{L}_{T,a^\star} + \sum_{t=1}^{T-1} \bar{F}_\eta^\star(-\hat{L}_t + \tilde{B}_t) - \bar{F}_\eta^\star(-\hat{L}_t + \tilde{B}_{t-1}).
$$

Note that $\bar{F}_\eta^\star(-\hat{L}_t + \tilde{B}_t) = \langle p_{t+1}, -\hat{L}_t + \tilde{B}_t \rangle + \eta^{-1} F(p_{t+1})$. Furthermore we have the bounds

$$
-\bar{F}_\eta^\star(-\hat{L}_T + \tilde{B}_{T-1}) \leq -\left(\langle \mathbf{e}_{a^\star}, -\hat{L}_T + \tilde{B}_{T-1} \rangle - \eta^{-1} F(\mathbf{e}_{a^\star})\right),
$$

and

$$
-\bar{F}_\eta^\star(-\hat{L}_t + \tilde{B}_{t-1}) \leq -\left(\langle p_{t+1}, -\hat{L}_t + \tilde{B}_{t-1} \rangle - \eta^{-1} F(p_{t+1})\right).
$$

Plugging these in leads to

$$\bar{F}^\star_\eta(0) - \bar{F}^\star_\eta(-\hat{L}_T + \tilde{B}_{T-1}) - \hat{L}_{T,a^\star} + \sum_{t=1}^{T-1} \bar{F}^\star_\eta(-\hat{L}_t + \tilde{B}_t) - \bar{F}^\star_\eta(-\hat{L}_t + \tilde{B}_{t-1})$$

$$\leq \frac{F(\mathbf{e}_{a^\star}) - F(p_1)}{\eta} - \tilde{B}_{T-1,a^\star} + \sum_{t=1}^{T-1} \langle p_{t+1}, \tilde{B}_t - \tilde{B}_{t-1} \rangle$$

$$\leq (\rho^\alpha_{1,a^\star} - \rho^\alpha_{T,a^\star})R + \frac{\mathrm{diam}(F)}{\eta} + \sum_{t=1}^{T-1} \langle p_{t+1}, B_t - B_{t-1} \rangle \,.$$

To bound the final sum above, note that for each coordinate $i$, the difference $B_{t,i} - B_{t-1,i}$ can be non-zero only if $p_{t+1,i}$ achieves $p_{t+1,i} = \rho^{-1}_{t+1,i}$. Therefore, we have

$$p_{t+1,i}(B_{t,i} - B_{t-1,i}) = R\rho^{-1}_{t+1,i} \left( \rho^\alpha_{t+1,i} - \rho^\alpha_{t,i} \right)$$

$$= \alpha R \int_{\rho_{t,i}}^{\rho_{t+1,i}} x^{\alpha-1} \rho^{-1}_{t+1} \, dx$$

$$\leq \alpha R \int_{\rho_{t,i}}^{\rho_{t+1,i}} x^{\alpha-2} \, dx$$

$$= \frac{\alpha R}{1-\alpha} (\rho^{\alpha-1}_{t,i} - \rho^{\alpha-1}_{t+1,i}) \,.$$

Applying this bound to each coordinate, we have

$$\sum_{t=1}^{T-1} \langle p_{t+1}, B_t - B_{t-1} \rangle = \sum_{i=1}^{M} \frac{\alpha R}{1-\alpha} \left( \rho^{\alpha-1}_{1,i} - \rho^{\alpha-1}_{T,i} \right) = \sum_{i=1}^{M} \frac{\alpha R}{1-\alpha} \left( \rho^{\alpha-1}_{1,i} - \rho^{\alpha-1}_{T,i} \right) \,.$$

Combining all of the bounds so far concludes the proof. $\qquad\square$

**Proof of Corollary 2.** The Tsallis regularizer is

$$F(x) = -\sum_{i=1}^{M} 2\sqrt{x_i} \,,$$

with a stability for the loss range $[0, L]$ of $L^2\sqrt{M}$ and a diameter of $2\sqrt{M}$[48][9]. Due to the symmetry of the potential, we have $\forall i : p_{1,i} = 1/M$. Using Theorem 6 with the loss range $[0, 2]$ leads to

$$\mathrm{Reg}_M(T) \leq 4\sqrt{2MT} + \left[ \frac{\alpha}{1-\alpha} \sum_{i=1}^{M} (M^{\alpha-1} - \mathbb{E}[\rho^{\alpha-1}_{T,i}]) + M^\alpha - \mathbb{E}[\rho^\alpha_{T,m^\star}] \right] R$$

$$\leq 4\sqrt{2MT} + \left[ \frac{\alpha}{1-\alpha} M^\alpha \left( 1 - M^{1-\alpha} \min_{j\in[M]} \mathbb{E}[\rho^{\alpha-1}_{T,j}] \right) + M^\alpha - \mathbb{E}[\rho^\alpha_{T,m^\star}] \right] R \,.$$

Dropping the negative $-M^{1-\alpha} \min_{j\in[M]} \mathbb{E}[\rho^{\alpha-1}_{T,j}]$ term leads to the first part of the $\min\{\cdot\}$ expression in Eq. (10). For the other term in the $\min\{\cdot\}$, note that the function

$$\alpha \mapsto \frac{\alpha}{1-\alpha} \left( 1 - z^{\alpha-1} \right)$$

is monotonically increasing in $\alpha$ with

$$\lim_{\alpha\to 1} \frac{\alpha}{1-\alpha} \left( 1 - z^{\alpha-1} \right) = \log(z) \,.$$

Absorbing $\log(\max_{j\in[M]} \mathbb{E}[\rho_{T,j}]/M) + 1$ by $2\log(\max_{j\in[M]} \mathbb{E}[\rho_{T,j}])$ (using that $\rho_{1,i} = M$) completes the proof. $\qquad\square$

# E  Approximation Algorithms for the Log-Determinant Barrier Problem

Recall that at every step, SquareCB.Inf (Algorithm 2) needs to sample from any distribution in logdet-barrier$(\hat{\theta}, \gamma; \mathcal{A})$, which is defined as

$$p^\star \in \underset{p \in \Delta(\mathcal{A})}{\operatorname{argmin}} \, \gamma \langle \bar{a}_p, \hat{\theta} \rangle - \frac{1}{\gamma} \log \det \left( H_p - \bar{a}_p \bar{a}_p^\top \right) , \tag{19}$$

where $\bar{a}_p = \mathbb{E}_{a \sim p}[a]$ and $H_p = \mathbb{E}_{a \sim p}[aa^\top]$. In this section, we develop optimization algorithms to efficiently find approximate solutions to the problem Eq. (19). In particular, our main result will be to prove Proposition 1.

While this is a convex optimization problem, developing efficient algorithms presents a number of technical difficulties. First, the optimization problem is non-smooth due to the presence of the log-determinant function, which prevents us from applying standard first-order methods such as gradient descent out of the box. Second, representing distributions in $\Delta(\mathcal{A})$ naively requires $\Omega(|\mathcal{A}|)$ memory. To get the result in Proposition 1, we employ a specialized Frank-Wolfe-type method, which maintains a sparse distribution and requires only $\mathcal{O}(\log|\mathcal{A}|)$ memory.

As a first step toward solving the problem numerically, we move to an equivalent but slightly more convenient formulation which lifts the actions to $d + 1$ dimensions. Define the *lifting* operator, which adds a new coordinate with 1 to each vector, by

$$\tilde{a} := \left( \begin{array}{c} a \\ 1 \end{array} \right) ,$$

and define

$$\tilde{a}_p := \mathbb{E}_{a \sim p}[\tilde{a}], \quad \tilde{H}_p := \mathbb{E}_{a \sim p}\big[\tilde{a}\tilde{a}^\top\big], \quad \tilde{\theta} := \left( \begin{array}{c} \hat{\theta} \\ 0 \end{array} \right), \quad \text{and} \quad \tilde{d} := d + 1 .$$

Furthermore, we define

$$G(p) = \langle \tilde{a}_p, \tilde{\theta} \rangle - \frac{1}{\gamma} \log \det(\tilde{H}_p). \tag{20}$$

**Proposition 2.** *The set of solutions for the lifted problem*

$$\underset{p \in \Delta(\mathcal{A})}{\operatorname{argmin}} \, G(p) = \underset{p \in \Delta(\mathcal{A})}{\operatorname{argmin}} \langle \tilde{a}_p, \tilde{\theta} \rangle - \frac{1}{\gamma} \log \det(\tilde{H}_p) , \tag{21}$$

*is identical to the set of solutions for Eq. (19), and vice-versa.*

**Proof.** By Lemma 4, any solution $p^\star$ to Eq. (19) must satisfy the optimality condition

$$\forall a \in \mathcal{A} \colon \langle \bar{a}_{p^\star} - a, \hat{\theta} \rangle + \frac{1}{\gamma} \| \bar{a}_{p^\star} - a \|^2_{(H_{p^\star} - \bar{a}_{p^\star} \bar{a}_{p^\star}^\top)^{-1}} \leq \frac{d}{\gamma} .$$

Now, let $p^\star$ be a minimizer for the optimization problem in (21). By first order optimality, we have

$$\forall p' \in \Delta(\mathcal{A}) \colon \sum_{a \in \operatorname{supp}(p^\star) \cup \operatorname{supp}(p')} (p'_a - p^\star_a) \left( \langle \tilde{a}, \tilde{\theta} \rangle - \frac{1}{\gamma} \| \tilde{a} \|^2_{\tilde{H}_{p^\star}^{-1}} \right) \geq 0 .$$

By the K.K.T. conditions, this condition holds if and only if there exists $\lambda \in \mathbb{R}$ such that

$$\forall a \in \operatorname{supp}(p^\star) \colon \langle \tilde{a}, \tilde{\theta} \rangle - \frac{1}{\gamma} \| \tilde{a} \|^2_{\tilde{H}_{p^\star}^{-1}} = \lambda \tag{22}$$

and

$$\forall a \in \mathcal{A} \colon \langle \tilde{a}, \tilde{\theta} \rangle - \frac{1}{\gamma} \| \tilde{a} \|^2_{\tilde{H}_{p^\star}^{-1}} \geq \lambda . \tag{23}$$

Note that Eq. (22) implies that

$$\mathbb{E}_{a \sim p^\star} \left[ \langle \tilde{a}, \tilde{\theta} \rangle - \frac{1}{\gamma} \| \tilde{a} \|^2_{\tilde{H}_{p^\star}^{-1}} \right] = \langle \tilde{a}_{p^\star}, \hat{\theta} \rangle - \frac{\tilde{d}}{\gamma} = \lambda .$$

Combining this identity with Eq. (23) and rearranging, we conclude that

$$\forall a \in \mathcal{A} : \langle \tilde{a}_{p^\star} - a, \hat{\theta} \rangle + \frac{1}{\gamma} \|\tilde{a}\|^2_{\tilde{H}_{p^\star}^{-1}} \le \frac{\tilde{d}}{\gamma}. \tag{24}$$

Finally, observe that for any $p \in \Delta(\mathcal{A})$

$$\tilde{H}_p = \begin{pmatrix} H_p & \bar{a}_p \\ \bar{a}_p^\top & 1 \end{pmatrix}, \quad \text{and} \quad \tilde{H}_p^{-1} = \begin{pmatrix} \left(H_p - \bar{a}_p \bar{a}_p^\top\right)^{-1} & -\left(H_p - \bar{a}_p \bar{a}_p^\top\right)^{-1} \bar{a}_p \\ -\bar{a}_p^\top \left(H_p - \bar{a}_p \bar{a}_p^\top\right)^{-1} & 1 + \|\bar{a}_p\|^2_{\left(H_p - \bar{a}_p \bar{a}_p^\top\right)^{-1}} \end{pmatrix},$$

where the second expression uses the identity for the Schur complement. Using the latter expression, we have that

$$\|\tilde{a}\|^2_{\tilde{H}_p^{-1}} = \|a\|^2_{(H_p - \bar{a}_p \bar{a}_p^\top)^{-1}} - 2a^\top \left(H_p - \bar{a}_p \bar{a}_p^\top\right)^{-1} \bar{a}_p + \|\bar{a}_p\|^2_{(H_p - \bar{a}_p \bar{a}_p^\top)^{-1}} + 1$$

$$= \|a - \bar{a}_p\|^2_{(H_p - \bar{a}_p \bar{a}_p^\top)^{-1}} + 1. \tag{25}$$

By plugging this expression into Eq. (24), it follows that the optimality conditions for the problems (21) and (19) are identical. Any solution $p^\star$ to the problem (21) yields a solution to the problem (19), and vice-versa. $\qquad\square$

In light of Proposition 2, we will work exclusively with the lifted problem going forward. Before stating our algorithm, we introduce the following approximate version of the optimality condition in Eq. (4), which quantifies the quality of a candidate solution $p \in \Delta(\mathcal{A})$.

**Definition 2.** For any action set $\mathcal{A}$, parameter $\hat{\theta} \in \mathbb{R}^d$ and learning rate $\gamma > 0$, a distribution $p \in \Delta(\mathcal{A})$ is called an $\eta$-*rounding* if it satisfies

$$\forall a \in \mathcal{A}: \quad \frac{1}{\gamma} \|\tilde{a}\|^2_{\tilde{H}_p^{-1}} \le (1 + \eta) \left( \frac{\tilde{d}}{\gamma} + \langle \tilde{a} - \tilde{a}_p, \tilde{\theta} \rangle \right). \tag{26}$$

The following lemma quantifies the loss in regret incurred by sampling from an $\eta$-rounding for the logdet-barrier objective rather than an exact solution.

**Lemma 5.** Suppose that for all steps $t$, we sample from an $\eta$-rounding for logdet-barrier$(\mathcal{A}_t, \hat{\theta}_t, \gamma/(1+\eta))$ within Algorithm 2. Then the bound from Lemma 4 will increase by at most a factor of $1 + 2\eta$.

Lemma 5 implies that to achieve the regret bound from Theorem 2 up to a factor of 2, it suffices to find a $1/2$-rounding.

**Proof.** We first prove an analogue of the inequality in Lemma 4. Let $t$ be fixed and abbreviate $\hat{\theta} \equiv \hat{\theta}_t$. Assume without loss of generality that $d = \dim(\mathcal{A}_t)$. For an $\eta$-rounding $p$ that satisfies Eq. (26) with learning rate $\gamma' := \gamma/(1 + \eta)$, by the identity (25), the following inequalities are equivalent:

$$\frac{1}{\gamma'} \|\tilde{a}\|^2_{\tilde{H}_p^{-1}} \le (1 + \eta) \left( \frac{\tilde{d}}{\gamma'} + \langle a - \bar{a}_p, \hat{\theta} \rangle \right)$$

$$\iff \frac{1 + \eta}{\gamma} \|\tilde{a}\|^2_{\tilde{H}_p^{-1}} \le (1 + \eta) \left( \frac{\tilde{d}(1 + \eta)}{\gamma} + \langle a - \bar{a}_p, \hat{\theta} \rangle \right)$$

$$\iff \frac{1}{\gamma} \left( \|a - \bar{a}_p\|^2_{(H_p - \bar{a}_p \bar{a}_p^\top)^{-1}} + 1 \right) \le \frac{(d + 1)(1 + \eta)}{\gamma} + \langle a - \bar{a}_p, \hat{\theta} \rangle$$

$$\iff \langle \bar{a}_p - a, \hat{\theta} \rangle + \frac{1}{\gamma} \|a - \bar{a}_p\|^2_{(H_p - \bar{a}_p \bar{a}_p^\top)^{-1}} \le \frac{d}{\gamma} \left( 1 + \eta + \frac{\eta}{d} \right).$$

It follows that the bound from Lemma 4 increases by at most a factor of $(1 + \eta + \frac{\eta}{d}) < 1 + 2\eta$ if we use an $\eta$-rounding rather than an exact solution. $\qquad\square$

### E.1 Algorithm

**Preliminaries.** To keep notation compact, throughout this section we drop the learning rate parameter and work with

$$G(p) := \langle \tilde{a}_p, \tilde{\theta} \rangle - \log \det(\tilde{H}_p), \quad \text{and} \quad p^\star \in \underset{p \in \Delta(\mathcal{A})}{\arg\min} \, G(p). \tag{27}$$

Note that this suffices to capture the case where $\gamma \neq 1$ (Eq. (20)), since we can multiply both sides by $\gamma$ and absorb a gamma factor into $\theta$. Consequently, for the remainder of the section we work under the assumption that $\|\theta\| \leq \gamma$ rather than $\|\theta\| \leq 1$. The definition of an $\eta$-rounding remains unaffected, since we can multiply both sides in Eq. (26) by $\gamma$.

**Additional notation.** For each $a \in \mathcal{A}$, let $\mathbf{e}_a \in \Delta(\mathcal{A})$ be the distribution that selects $a$ with probability 1. For any distributions $p_1, p_2 \in \Delta(\mathcal{A})$, let $\text{conv}[p_1, p_2] = \{\lambda p_1 + (1 - \lambda)p_2 \mid \lambda \in [0, 1]\}$ be their convex hull. To improve readability, we abbreviate $\|\cdot\|_{\tilde{H}_p^{-1}}$ to $\|\cdot\|_p$ in this section.

**Algorithm.** Our main algorithm is stated in Algorithm 6. The algorithm is a generalization of Khachiyan's algorithm for optimal design [30]. It maintains a finitely supported distribution over arms in $\mathcal{A}$ and adds a single arm to the support at each step.

In more detail, the algorithm proceeds as follows. At step $k$, the algorithm checks whether the current iterate $p_{k-1}$ is an $\eta$-rounding. If this is the case, the algorithm simply terminates, as we are done. Otherwise, with $a^\star := \arg\min_{a \in \mathcal{A}} \langle a, \theta \rangle$, the algorithm first checks whether the current distribution satisfies $\tilde{d} + \langle a^\star - \bar{a}_{p_{k-1}}, \theta \rangle \geq 1$. If that condition is violated, we define a new distribution $p'_{k-1}$ by choosing the distribution in $\text{conv}[p_{k-1}, e_{a^\star}]$ that minimizes $G(p)$. This ensures that $\frac{\partial}{\partial \lambda}[G(p'_{k-1} + x(\mathbf{e}_{a^\star} - p_{k-1})](0) = 0$ the same as the one along $p'_{k-1}$, i.e.

$$\langle a^\star, \theta \rangle - \|a^\star\|_{p_{k-1}}^2 = \mathbb{E}_{a \sim p'_{k-1}}\left[\langle a, \theta \rangle - \|a\|_{p_{k-1}}^2\right] = \langle \bar{a}_{p'_{k-1}}, \theta \rangle - \tilde{d},$$

and hence $\min_{a \in \mathcal{A}} \tilde{d} + \langle a - \bar{a}_{p'_{k-1}}, \theta \rangle = \|a^\star\|_{p_{k-1}}^2 \geq 1$. This ensures in particular that

$$\eta_k := \max_{a \in \mathcal{A}} \|\tilde{a}\|_{p'_{k-1}}^2 / (d + \langle a - \bar{a}_{p'_{k-1}}, \theta \rangle) \tag{28}$$

is well defined. To conclude the iteration, the algorithm selects an action $a_k$ that attains the maximum in Eq. (28) and adds it to the support of $p'_{k-1}$, yielding $p_k$.

---

**Algorithm 6:** Frank-Wolfe for minimizing the logdet-barrier objective

---

**Input:** $p_0 \in \Delta(\mathcal{A}), \mathcal{A}, \theta, \eta$
Let $a^\star = \arg\min_{a \in \mathcal{A}} \langle a, \theta \rangle$, $k = 1$.
**while** $p_{k-1}$ is not an $\eta$-rounding (Eq. (26)) **do**
  **if** $\tilde{d} + \langle a^\star - \bar{a}_{p_{k-1}}, \theta \rangle < 1$ **then**
  $\quad \lfloor$ Solve $p'_{k-1} = \arg\min_{p \in \text{conv}[p_{k-1}, \mathbf{e}_{a^\star}]} G(p)$.
  **else**
  $\quad \lfloor$ $p'_{k-1} = p_{k-1}$.
  Pick any $a_k \in \arg\max \|\tilde{a}\|_{p'_{k-1}}^2 / (d + \langle a - \bar{a}_{p'_{k-1}}, \theta \rangle)$ (ties broken arbitrarily).
  Solve $p_k = \arg\min_{p \in \text{conv}[p'_{k-1}, \mathbf{e}_{a_k}]} G(p)$.
  Increment $k$.

---

### E.2 Analysis

In this section we prove a number of intermediate results used to bound the iteration complexity of Algorithm 6, culminating in our main convergence guarantee, Theorem 7. The total computational complexity is summarized at the end of the section in Appendix E.2.1.

We begin by relating the $\eta$-rounding property to the suboptimality gap for the objective $G(p)$.

**Lemma 6.** If $p \in \Delta(\mathcal{A})$ is an $\eta$-rounding, then

$$G(p) - G(p^\star) \leq \log(1 + \eta)\tilde{d}.$$

**Proof of Lemma 6.** By the optimality conditions in Eqs. (22) to (24), we are guaranteed that

$$\forall a \in \operatorname{supp}(p^\star) : \tilde{d} + \langle a, \theta \rangle = \|\tilde{a}\|^2_{\tilde{H}^{-1}_{p^\star}} + \langle \bar{a}_{p^\star}, \theta \rangle .$$

Hence, combining this statement with the $\eta$-rounding condition for $p$, we have that

$$\forall a \in \operatorname{supp}(p^\star) : \|\tilde{a}\|^2_{\tilde{H}^{-1}_p} \leq (1+\eta) \left( \|\tilde{a}\|^2_{\tilde{H}^{-1}_{p^\star}} + \langle \bar{a}_{p^\star} - \bar{a}_p, \theta \rangle \right) .$$

Taking the expectation over $a \sim p^\star$ on both sides above and rearranging leads to

$$\langle \bar{a}_p - \bar{a}_{p^\star}, \theta \rangle \leq \tilde{d} - \frac{\operatorname{tr}(\tilde{H}_{p^\star} \tilde{H}^{-1}_p)}{1+\eta} = \tilde{d} - \frac{\operatorname{tr}(\tilde{H}^{\frac{1}{2}}_{p^\star} \tilde{H}^{-1}_p \tilde{H}^{\frac{1}{2}}_{p^\star})}{1+\eta} .$$

From the definition of $G(p)$, this implies that

$$G(p) - G(p^\star) \leq \tilde{d} - \frac{\operatorname{tr}(\tilde{H}^{\frac{1}{2}}_{p^\star} \tilde{H}^{-1}_p \tilde{H}^{\frac{1}{2}}_{p^\star})}{1+\eta} + \log \det(\tilde{H}^{\frac{1}{2}}_{p^\star} \tilde{H}^{-1}_p \tilde{H}^{\frac{1}{2}}_{p^\star}),$$

where we recall that $\det(\tilde{H}^{\frac{1}{2}}_{p^\star} \tilde{H}^{-1}_p \tilde{H}^{\frac{1}{2}}_{p^\star}) = \det(\tilde{H}_{p^\star} \tilde{H}^{-1}_p) > 0$, since $\tilde{H}_{p^\star}, \tilde{H}_p \succ 0$. Now, let $(\lambda_i)_{i=1,\ldots,\tilde{d}}$ be the eigenvalues of $\tilde{H}^{\frac{1}{2}}_{p^\star} \tilde{H}^{-1}_p \tilde{H}^{\frac{1}{2}}_{p^\star}$. Then we have

$$G(p) - G(p^\star) = \sum_{i=1}^{\tilde{d}} 1 - \frac{\lambda_i}{1+\eta} + \log(\lambda_i) \leq \tilde{d} \max_{\lambda > 0} \left\{ 1 - \frac{\lambda}{1+\eta} + \log(\lambda) \right\} = \tilde{d} \log(1+\eta) .$$

$\square$

Our next lemma lower bounds the rate at which the suboptimality gap improves at each iteration.

**Lemma 7.** In each iteration of Algorithm 6, the suboptimality gap improves by at least

$$G(p_{k-1}) - G(p_k) \geq \Omega \left( \min\{\eta_k, 1\}^2 / d \right), \tag{29}$$

where we recall that $\eta_k := \|a_k\|^2_{p'_{k-1}} / (\tilde{d} + \langle a_k - \bar{a}_{p'_{k-1}} \rangle)$. Furthermore, if $\eta_k \geq 2\tilde{d}$, then it also holds that

$$G(p_k) - G(p^\star) \leq \left( 1 - \frac{1}{2\tilde{d}} \right) (G(p_{k-1}) - G(p^\star)) . \tag{30}$$

**Proof.** We first prove that Eq. (29) holds. Let $k$ be fixed, and let $\alpha \in [0,1]$ such that $p_k = (1-\alpha)p'_{k-1} + \alpha \mathbf{e}_{a_k}$. Then we have

$$G(p_k) = \langle \bar{a}_{p_k}, \theta \rangle - \log \det \left( \tilde{H}_{p_k} \right)$$

$$= (1-\alpha)\langle \bar{a}_{p'_{k-1}}, \theta \rangle + \alpha\langle \tilde{a}_k, \theta \rangle - \log \det \left( (1-\alpha)\tilde{H}_{p'_{k-1}} + \alpha \tilde{a}_k \tilde{a}_k^\top \right)$$

$$= \langle \bar{a}_{p'_{k-1}}, \theta \rangle + \alpha\langle \tilde{a}_k - \bar{a}_{p'_{k-1}}, \theta \rangle - \log \left( \det \left( (1-\alpha)\tilde{H}_{p'_{k-1}} \right) \cdot \left( 1 + \frac{\alpha}{1-\alpha} \|\tilde{a}_k\|^2_{p'_{k-1}} \right) \right)$$

$$= G(p'_{k-1}) + \alpha\langle \tilde{a}_k - \bar{a}_{p'_{k-1}}, \theta \rangle - (\tilde{d}-1) \log(1-\alpha) - \log \left( 1 - \alpha + \alpha \|\tilde{a}_k\|^2_{p'_{k-1}} \right),$$

where the third equality uses the matrix determinant lemma. Now, recall that by the definition of $a_k$, we have $\|\tilde{a}_k\|^2_{p'_{k-1}} = (1+\eta_k)(\tilde{d} + \langle \tilde{a}_k - \bar{a}_{p'_{k-1}}, \theta \rangle)$. Let us abbreviate $Z_k := \|\tilde{a}_k\|^2_{p'_{k-1}} \geq 1 + \eta_k$. We proceed as

$$G(p_{k-1}) - G(p_k) \geq G(p'_{k-1}) - G(p_k)$$

$$= \alpha\langle \bar{a}_{p'_{k-1}} - \tilde{a}_k, \theta \rangle + (\tilde{d}-1)\log(1-\alpha) + \log \left( 1 - \alpha + \alpha \|\tilde{a}_k\|^2_{p'_{k-1}} \right)$$

$$= \alpha \left( \tilde{d} - \frac{Z_k}{1+\eta_k} \right) + (\tilde{d}-1)\log(1-\alpha) + \log \left( 1 + \alpha(Z_k - 1) \right)$$

$$= \max_{\alpha' \in [0,1]} \left\{ \alpha' \left( \tilde{d} - \frac{Z_k}{1+\eta_k} \right) + (\tilde{d}-1)\log(1-\alpha') + \log \left( 1 + \alpha'(Z_k - 1) \right) \right\}, \tag{31}$$

where the last equality uses that $\alpha$ is chosen such that $G(p_k)$ is minimized. Next, recalling the elementary fact that for all $x \geq -\frac{1}{2}$, $\log(1+x) \geq x - x^2$, we have in particular that

$$G(p_{k-1}) - G(p_k)$$

$$\geq \max_{\alpha' \geq \frac{1}{2}} \left\{ \alpha' \left( \tilde{d} - \frac{Z_k}{1+\eta_k} \right) + (\tilde{d}-1)(-\alpha' - \alpha'^2) + \alpha'(Z_k - 1) - \alpha'^2 (Z_k - 1)^2 \right\}$$

$$= \max_{\alpha' \geq \frac{1}{2}} \left\{ \alpha' \frac{\eta_k Z_k}{1+\eta_k} - \alpha'^2 \left( \tilde{d} - 1 + (Z_k - 1)^2 \right) \right\}.$$

Note that $\tilde{d} \geq 3$ and $\max_{x>0} \frac{x}{2+(x-1)^2} \leq 1$, so if we choose

$$\alpha' = \frac{\eta_k Z_k}{2(1+\eta_k)\left(\tilde{d} - 1 + (Z_k - 1)^2\right)} \leq \frac{1}{2},$$

we get the lower bound

$$G(p_{k-1}) - G(p_k) \geq \frac{\eta_k^2 Z_k^4}{4(1+\eta_k)^2 \left(\tilde{d} - 1 + (Z_k - 1)^2\right)}.$$

The proof of Eq. (29) now follows by noting that $\frac{x^2}{d+(x-1)^2} \geq \frac{1}{d}$ for all $x \geq 1$.

We now prove that the second part of the lemma, Eq. (30), holds. Suppose $\eta_k > 2\tilde{d}$. We return to Eq. (31) and this time select

$$\alpha' = \frac{\sqrt{\eta_k}}{Z_k - 1} \leq \frac{1}{\sqrt{\eta_k}} \leq \frac{1}{2}.$$

Using the approximation $\log(1+x) \geq x - x^2$ only for the first term in (31), we get

$$G(p_{k-1}) - G(p_k) \geq \alpha' \left( \tilde{d} - \frac{Z_k}{1+\eta_k} \right) - (\tilde{d}-1)(\alpha' + \alpha'^2) + \log\left(1 + \alpha'(Z_k - 1)\right)$$

$$\geq -\frac{\sqrt{\eta_k}}{1+\eta_k} - \frac{\tilde{d}-1}{\eta_k} + \log(1 + \sqrt{\eta_k})$$

$$= -\frac{\sqrt{\eta_k}}{1+\eta_k} - \frac{\tilde{d}-1}{\eta_k} + \log(1 + \sqrt{\eta_k}) - \frac{1}{4}\log(1+\eta_k) + \frac{1}{4}\log(1+\eta_k)$$

$$\geq -\frac{\sqrt{\eta_k}}{1+\eta_k} - \frac{1}{2} + \frac{1}{\eta_k} + \log(1 + \sqrt{\eta_k}) - \frac{1}{4}\log(1+\eta_k) + \frac{1}{4}\log(1+\eta_k),$$

where the last line uses that $\eta_k \geq 2\tilde{d}$. Now observe that for $x \geq 6$

$$\frac{\partial}{\partial x} \left( -\frac{\sqrt{x}}{1+x} + \frac{1}{x} + \log(1 + \sqrt{x}) - \frac{1}{4}\log(1+x) \right)$$

$$= \frac{x-1}{2\sqrt{x}(1+x)^2} - \frac{1}{x^2} + \frac{1}{2(\sqrt{x}+x)} - \frac{1}{4(1+x)}$$

$$= \frac{x^{\frac{7}{2}} + x^3 + 5x^{\frac{5}{2}} - 7x^2 - 12x^{\frac{3}{2}} - 8x - 4x^{\frac{1}{2}} - 4}{4x^2(1+\sqrt{x})(1+x)^2}$$

$$\geq \frac{7x^2 + 60x^{\frac{3}{2}} - 7x^2 - 12x^{\frac{3}{2}} - 8x - 4x^{\frac{1}{2}} - 4}{4x^2(1+\sqrt{x})(1+x)^2} \geq 0.$$

Hence

$$-\frac{\sqrt{\eta_k}}{1+\eta_k} - \frac{1}{2} + \frac{1}{\eta_k} + \log(1 + \sqrt{\eta_k}) - \frac{1}{4}\log(1+\eta_k)$$

$$\geq -\frac{\sqrt{6}}{1+6} - \frac{1}{2} + \frac{1}{6} + \log(1 + \sqrt{6}) - \frac{1}{4}\log(1+6) > 0.$$

It follows that

$$G(p_{k-1}) - G(p_k) \geq \frac{1}{4}\log(1+\eta_k).$$

$\square$

The next lemma ensures we can efficiently find a good initial distribution $p_0$.

**Lemma 8** (Kumar and Yildirim [32], Lemma 3.1). There exists an algorithm that terminates in $\mathcal{O}(|\mathcal{A}|d^2)$ time and finds a distribution $p_0 \in \Delta(\mathcal{A})$ with support $|\operatorname{supp}(p_0)| \leq 2\tilde{d}$ such that

$$-\log \det(\tilde{H}_{p_0}) + \min_{p \in \Delta(\mathcal{A})} \log \det(\tilde{H}_p) = \mathcal{O}(d \log(d)).$$

The memory requirement of this routine is $\mathcal{O}\left(d^2 + \log(|\mathcal{A}|d)\right)$.

**Corollary 4.** *The distribution of Lemma 8 has an initial suboptimality gap of*

$$G(p_0) - G(p^\star) = \mathcal{O}(d \log(d) + \gamma).$$

**Proof.** Recall that

$$G(p_0) - G(p^\star) = \langle \bar{a}_{p_0} - \bar{a}_{p^\star}, \theta \rangle - \log \det(\tilde{H}_{p_0}) + \log \det(\tilde{H}_{p^\star}).$$

The difference between the log-det terms is bounded by $\mathcal{O}(d \log(d))$ using Lemma 8, while the difference between the linear terms is bounded by

$$\langle \bar{a}_{p_0} - \bar{a}_{p^\star}, \theta \rangle \leq \|\bar{a}_{p_0} - \bar{a}_{p^\star}\| \cdot \|\theta\| \leq 2\gamma.$$

$\square$

**Theorem 7.** *If Algorithm 6 is initialized using the distribution of Lemma 8, then it requires $\mathcal{O}(d(\log(d) + \log(\gamma)))$ iterations to reach a $2d$-rounding. Moreover,*

- *After reaching the $2d$-rounding above, the algorithm requires $\mathcal{O}(\log(d)d^2)$ additional iterations to reach a $1$-rounding.*

- *After reaching such a $1$-rounding, the algorithm requires $\mathcal{O}(d^2/\eta)$ additional iterations to reach an $\eta$-rounding for any $\eta < 1$.*

*Altogether, for any $\eta > 0$, Algorithm 6—when initialized using Lemma 8—requires*

$$\mathcal{O}(d \log(\gamma) + d^2(\log(d) + 1/\eta))$$

*total steps to reach an $\eta$-rounding.*

**Proof.** By Corollary 4 we know that the initial distribution $p_0$ satisfies

$$G_0 := G(p_0) - G(p^\star) = \mathcal{O}(d \log(d) + \gamma).$$

We first consider bound the number of steps required to reach a $2d$-rounding. Let $k_0$ denote the first step $k$ in which $p_k$ is a $2d$-rounding. Then every $k < k_0$ has $\eta_k > 2d$, so in light of Lemma 7, all such $k$ have

$$G(p_k) - G(p_0) \leq \left(1 - \frac{1}{2\tilde{d}}\right)(G(p_{k-1}) - G(p_0))$$

and

$$G(p_k) \leq G(p_{k-1}) - \Omega(1/d).$$

It follows that as long as $\eta_k > 2d$, the suboptimality gap will reach $1$ in most $\mathcal{O}(d \log(G_0)) = \mathcal{O}(d(\log(d) + \log(\gamma)))$ iterations. Moreover, since the absolute decrease in function value is at least $\Omega(1/d)$, the gap will reach zero after another $\mathcal{O}(d)$ iterations. We conclude that after $\mathcal{O}(d(\log(d) + \log(\gamma)))$ iterations, the algorithm must find a $2d$-rounding.

We now bound the number of steps to reach a $1$-rounding from the first step where we have a $2d$-rounding. By Lemma 6, the suboptimality gap of any $2d$-rounding is at most $\mathcal{O}(d \log(d))$. Moreover, as long as we haven't reached a $1$-rounding, Lemma 7 guarantees that the suboptimality gap will decrease by $\Omega(1/d)$ per step. Hence, we must reach a $1$-rounding within $\mathcal{O}(d^2 \log(d))$ iterations.

Finally we bound the number of steps required to reach an $\eta$-rounding for any $\eta < 1$, starting from the first iteration where we reach a $1$-rounding. We adapt an argument of Kumar and Yildirim [32]. Given an $\eta_k$-rounding for $\eta_k \leq 1$, we need $\mathcal{O}(d^2/\eta_k)$ iterations to reach an $(\eta_k/2)$-rounding. This follows from the same argument as above: the suboptimality gap is at most $\mathcal{O}(d\eta_k)$ by Lemma 6 (using that $\log(1 + \eta_k) \leq \eta_k$) and we reduce it by $\Omega(\eta_k^2/d)$ as long as we have not found an $(\eta_k/2)$-rounding (by Lemma 7). Summing up the required number of iterations to get from precision $1$ to $1/2$ to $1/4$ to ... to $1/2^{\lceil \log_2(1/\eta) \rceil}$ shows that $\mathcal{O}(d^2/\eta)$ total iterations suffice. $\square$

### E.2.1    Total Computational Complexity

The computational complexity per iteration for our method is comparable to similar algorithms for the D-optimal design problem (the case $\theta = 0$) [30, 32, 44]. We walk through the computation complexity step-by-step for completeness, and to handle differences arising from our generalization to the $\theta \neq 0$ case.

The first difference is that we have an intermediate optimization along the line $\mathrm{conv}(p_{k-1}, \mathbf{e}_{a^\star})$. This step increases the computational complexity by a factor of 2. At each iteration, Algorithm 6 computes

$$\operatorname*{argmax}_{a \in \mathcal{A}} \frac{\|\tilde{a}\|^2_{p'_{k-1}}}{d + \langle a - \bar{a}_{p'_{k-1}}, \theta \rangle} \, .$$

For generic action sets, this can be computed in time $\mathcal{O}(|\mathcal{A}|d^2)$, given that $\tilde{H}^{-1}_{p'_{k-1}}$ has already been computed.

In the next step, the algorithm solves the one dimensional optimization problem

$$\max_{\alpha' \in [0,1]} \left( \alpha' \left( \tilde{d} - \frac{Z_k}{1 + \eta_k} \right) + (\tilde{d} - 1) \log(1 - \alpha') + \log\left(1 + \alpha'(Z_k - 1)\right) \right) \, ,$$

where $Z_k = \|\tilde{a}_k\|^2_{p'_{k-1}}$. This can be done in time $\mathcal{O}(1)$, since it is equivalent to solving the quadratic problem

$$\left( \tilde{d} - \frac{Z_k}{1 + \eta_k} \right) - \frac{\tilde{d} - 1}{1 - x} + \frac{Z_k - 1}{1 + x(Z_k - 1)} = 0 \, .$$

Finally we need to update $\bar{a}_p$, which costs $\mathcal{O}(d)$, and update $\tilde{H}^{-1}_p$, which can be done in time $\mathcal{O}(d^2)$ using a rank-one update.

Across all iterations, we require a total of $\tilde{\mathcal{O}}(d^4|\mathcal{A}|)$ arithmetic operations, with $p_k$ never exceeding a support of $\mathcal{O}\big(d^2 \log(d) + d \log(\gamma)\big)$, since we maximally add one arm to the support in any iteration. We can store $p_k$ as a sparse vector of key and value pairs, where each entry has a memory complexity of $\mathcal{O}(\log(|\mathcal{A}|))$ to represent the keys.