[Reviews · NeurIPS 2020]

Review 1

Summary and Contributions: This paper studies Linear Contextual Bandit problem with context and set of available actions, generated by oblivious adversary, and a fixed loss functions. This is the first work, that considers infinite action set for this setting. The main focus of this work is on the misspecified case with unknown upper bound on the misspecification, that wasn’t studied before. The solution of this problem is based on the corralling procedure. Other contribution of this work is replacing dimension d in epsilon \sqrt{d} T term in the regret to the effective dimension of the sequence of sets of available actions.

Strengths: I find presented corralling procedure potentially useful for future works, like model selection. The result on the replacing dimension with effective dimension gives a big improvement for sparse problems.

Weaknesses: Optimisation problem in Definition 7 doesn’t look computationally efficient. Could you add more details on how do you propose to solve this problem?

Correctness: The analysis looks correct to me.

Clarity: Some problems with numeration of theorems, assumptions, etc. Apart from that the paper is well written.

Relation to Prior Work: Foster and Rakhlin, 2020 shows epsilon\sqrt{K}T term in the misspecification, which can be better than effective demention in many cases. Is it possible to get min(K, d) in your work?

Reproducibility: Yes

Additional Feedback:


Review 2

Summary and Contributions: This paper studies the misspecification issue for linear contextual bandits with a regression oracle. Specifically, built on [21], the authors extend the algorithm and the analysis to address the challenges of infinite action sets and the unknown misspecification level of the function class. The main contributions of this paper are three-fold: (i) It proposes to use the logdet-barrier distribution to address the infinite action sets; (ii) A new master algorithm is proposed to combine base bandit algorithms of different misspecification levels; (iii) It presents rigorous regret analysis of the proposed algorithms. ===== Post-Rebuttal ===== The authors' response addressed the major concerns. I have updated the score accordingly.

Strengths: 1. The bandit setting considered in this paper is important. 2. The proposed base algorithm ImpSquareCB and the master algorithm (\alpha, R)-hedged FTRL are novel. 3. The regret analysis is rigorous and non-trivial.

Weaknesses: 1. The presentation can be improved: - Several designs are introduced without much intuition. For example, in Algorithm 2, why is logdet-barrier a proper distribution for selecting the actions? What’s the intuition behind the design of the learning rate $\gamma_t$? - There is a big leap from Section 3.2 to Section 3.3. First, it is not well explained why the concept of importance-weighted regret is needed or why a master algorithm is required to address unknown misspecification. Section 3.3 is presented in a bottom-up manner and turns out to be a bit confusing. - In Lines 166-167, it is mentioned that adapting SquareCB to the misspecification setting is non-trivial, but without any explanation. 2. The lack of empirical results: While I usually do not comment on the lack of empirical results, for this topic a few sets of empirical results (e.g. synthetic experiments) will be very helpful in demonstrating the performance gain of using the proposed method, compared to other benchmarks that require either a realizability assumption or the knowledge of misspecification level.

Correctness: I have read through the proofs up to Section 4 and had a quick look at the proofs in Sections 5. While I do not find any specific errors in the proofs, there are indeed several equations that can be made more transparent (please see detailed comments below).

Clarity: Overall this paper is clearly written, except for Section 3.3 as pointed out above.

Relation to Prior Work: This paper has correctly cited the prior works and included all the important references.

Reproducibility: Yes

Additional Feedback: Some additional comments: Lines 164-165: The first claim in Remark 9 is presented without proof. This could be made more transparent at least by providing a proof sketch. Lines 183-184: Theorem 10 seems to presume that the function class consists of constant vector-valued functions based on the proof in Appendix C (Lines 486-487). However, from the description of Theorem 10, it is unclear what function class is considered. Line 205: This equation of regret is claimed but with its proof postponed to Line 500. It would be helpful to mention this right after Line 205. Line 451: How to obtain the last two equalities? In the second last equality, why is the expectation independent of $\hat{\theta}$? Please make this more transparent. Algorithms 1 and 2: The term “SqAlg” is not defined. Line 487: Regarding Reg_{Imp}, how to obtain the second equation from the first one?


Review 3

Summary and Contributions: This paper tackles the \varepsilon misspecified bandits without prior knowledge about misspecification level for infinite arms. With a regression oracle, optimal regret bound is proved. Moreover, they have derived a class of improved master algorithms for corralling. The reviewer likes the discussion about connection with adversarial corrupted bandits.

Strengths: The proofs and claims are sound. Although the topic related to misspecified bandit is not a new problem and has been well investigated, this paper is still worthwhile in some perspectives, e.g., the open problems from [27] with unknown \varepsilon and the improved master algorithms for corralling. This paper is relevance to the NeurIPS community.

Weaknesses: (1) The regret bound is still linear with T unless more assumption about \epsilon. Does this mean in practice misspecification will heavily harm the performance (always negative result)? This is a little different from other misspecified bandits. Could the authors give a non-regret setting ? e.g., the small deviation case of [22].

Correctness: The claims and method are correct.

Clarity: This paper is well written.

Relation to Prior Work: This paper clearly discussed how this work differs from previous contributions.

Reproducibility: Yes

Additional Feedback:


Review 4

Summary and Contributions: The paper considers the misspecified linear contextual bandit problem and proposes a novel contextual bandit algorithm robust to misspecification. The algorithm relies on calls to an optimal online regression oracle, and when specialized to linear contextual bandits with infinite actions achieves optimal regret guarantees that decompose additively into two terms, one of them depending linearly on the misspecification parameter. Moreover, the algorithm does not need to know the true misspecification parameter but can adapt to misspecification.

Strengths: This paper solves an interesting open problem posed in the work of Lattimore et al., for which the previous phased elimination type algorithms do not easily extend. The idea of this work is to consider another approach inspired by Foster and Rakhlin, and their idea of the reduction of contextual bandit to online regression. One of the main ideas in this paper of replacing abe-long with log-barrier (which allows them to handle infinite action sets) seems novel. Further on, non-trivial work is required to obtain the regret bound that matches the existing lower one. In particular, a novel algorithm based on Corraling + weighted modification of SquareCB is proposed. Finally, model mismatch in bandits and RL is a very important problem that has recently received a lot of attention. Hence, I think that this paper meets the requirements when it comes to both novelty and significance.

Weaknesses: I did not spot any major limitations/weaknesses in this work. However, the authors state that the algorithm is suitable for practical deployment, and although this a theoretical work, simulations/experiments that support this claim would be a plus. Also related to this, the overall computational complexity of the proposed method seems not discussed.

Correctness: From a limited inspection, the claims seem to be correct. The statement of Theorem 12 contains an additional sqrt{d} in the first term. By inspecting its proof, this seems to be a typo.

Clarity: Overall, the paper is well-written. Some suggestions for improvement are outlined in the "additional feedback" section.

Relation to Prior Work: I think that the authors have clearly explained the previous work and differences concerning the previous works of Foster and Rakhlin and SquareCB, and the work of Agarwal et al. They have also used quite some space in the main body of the paper to explain these methods which definitely improves readability. The difference with respect to the work of Lattimore et al. is clearly explained and some further related work on the adversarially corrupted bandits is provided.

Reproducibility: Yes

Additional Feedback: ________________________________ Post-rebuttal comments: After reading the rebuttal and other reviews, I keep my score. The main theoretical result is both novel and strong. There are some good suggestions when it comes to improving the overall clarity of the paper in the reviews and I hope the authors would consider them in the revised version. ________________________________ When explaining Algorithm 3, it is not clear how q_t gets updated (also in the provided pseudocode). This is only discussed later on in the paper. Perhaps, provide a reference to the section where this is discussed. Also, the paper would further benefit from more intution/discussion on the need for weighted regression oracle. Can you also comment on the computational complexity of the overall method? Can you explain the sampling procedure in Alg.2, i.e., a_t ~logdet-barrier, and the need of solving the problem in Definition 7 (i.e., its solution and method that solves it)? I’m curious about Assumption 1 in other settings rather than linear, e.g., the mentioned kernelized setting? Can you perhaps comment on the extension of your result to this setting? Does it make sense to talk about misspecification setting in the case of universal kernels such as, e.g., Gaussian kernel? Minor comments/typos: - Eq. 2, perhaps A_t instead of A(x_t) - 135, comma is missing {e1, …, ek} - Alg. 1, gamma is not defined/mentioned earlier - Eq. 3, both i^* and a^* are used. - 162, 2 x “the” - SqAlg is not defined - 172, here you talk about Bernoulli and q_t is its parameter, later on (e.g., 197) A_t ~q_t. - Algorithm 3, rho_t,A_t also needs to be passed - 201, empty space

[Author Response · NeurIPS 2020]

We thank all reviewers for their detailed feedback. We will be sure to address all questions and incorporate all
suggestions from the reviewers in the final version of the paper. **Note: Reference numbers below refer to the main**
**submission, not the supplementary version.** To reiterate and clarify, our contributions include the following:

1. We reduce contextual linear bandits with infinite action sets to a regression problem with an algorithm that is both
practical and efficient.
2. We optimally adapt to an unknown level of misspecification, which is a non-trivial open problem [27]. Recall
that previous works in contextual bandits required oracle knowledge of the misspecification level in order to tune the
algorithm's parameters (e.g., the upper confidence bound in LinUCB). Let us emphasize that neither doubling tricks
nor other methods were known to circumvent this requirement. The solution for static action sets heavily relies on
elimination, and does not generalize to the contextual case [27].
3. We adapt to a compelling notion of sparsity defined by an average effective dimension.
4. We provide a novel view of corralling bandits and give an improved master algorithm.

**Questions common to multiple reviewers**
- *Optimization problem:* Several valid questions were raised regarding solving the optimization problem in Def. 7 and
the associated computational cost. This is a convex optimization problem over a convex set with easily computable
gradients. Finding an $\epsilon$-approximation takes $\mathcal{O}(\mathrm{Poly}(d)\log(1/\epsilon))$ time. (see "Relatively-Smooth Convex Optimization
by First-Order Methods, and Applications" (Hu, Freund, Nesterov), Theorem 3.1). See also the optimal design example
in Sect. 2.2 therein which, up to a linear term and a benign term in the Hessian, is equivalent to our problem. At every
inner iteration, the solver needs to find $\mathrm{argmax}_{a\in\mathcal{A}_t}\langle a,\mu\rangle + \beta||a||^2_{H^{-1}}$, where $H^{-1}$ can be updated in $O(d^2)$ time,
while finding the argmax is the same problem that the standard LinUCB algorithm is solving. We will add a formal
proof and discussion along these lines to the paper.
- *Experiments:* We agree that experiments on real-world or synthetic data would be a bonus here, but we believe that our
strong theoretical results stand for themselves.

**Reviewer 1**
- *Foster and Rakhlin show ... can you get $min(K,d)$ ?* The effective dimension we use in Theorem 13 (L. 249) is upper
bounded by $K$, since the linear subspace spanned by the feature vectors of all arms trivially includes the action set. In
fact, when $K$ is equal to the effective dimension, then the logdet barrier and the logbarrier coincide.

**Reviewer 2**
- *On logdet-barrier being a proper distribution:* This holds by definition, because the optimization problem in Def. 7 is
over the probability simplex.
- *Adapting SquareCB to the misspecification setting is non-trivial:* Briefly, the optimal setting for the parameter $\gamma$ in
SquareCB (see Theorem 5/6 in [21]) depends on $\varepsilon$. Any choice for $\gamma$ that ignores $\varepsilon$ leads to suboptimal regret (e.g.,
using the optimal choice for $\varepsilon = 0$ leads to regret scaling as $\varepsilon^2 T^{3/2}$ when $\varepsilon \neq 0$). Hence, the purpose of the master
algorithm is to learn a near-optimal choice for $\gamma$ on-the-fly. See also Item 2 at the top of this page.
- *On assumptions in Theorem 10:* Theorem 10 is stated and proven for general function classes $f\in\mathcal{F}, f:\mathcal{X}\to\mathbb{R}^d$,
we don't see any inconsistency with LL. 486-487 (note that the $\theta_t^\star$ notation in this proof is just shorthand for $f^\star(x_t)$).
- *Line 487, how to obtain the 2nd eq. from the 1st one:* This follows by adding and subtracting terms, and then applying
the triangle inequality to term differences.

**Reviewer 3**
- *This is a little different ... the small deviation case of [22]:* These results are not comparable because their work only
considers fixed action sets. We are not aware of a suitable definition of "small deviation" for the contextual case with
changing action sets. We agree that it is an interesting direction for future research.
- *Regret bound still linear in $T$... :* There is a tight lower bound (see, e.g., [27]) showing that the $\varepsilon\sqrt{d}T$ term is generally
unavoidable, even if $\varepsilon$ is known beforehand. Nonetheless, notice that our bounds rely on the *empirical* quantity $\varepsilon_T \leq \varepsilon$
and, in practice, one may hope for an $\varepsilon_T$ of order $T^{-\alpha}$, for some positive $\alpha$, leading to no-regret results.

**Reviewer 4**
- *The statement of Theorem 12 contains an additional $\sqrt{d}$ ... seems to be a typo:* Indeed, thanks for spotting this!
- *On Assumption 1 in other settings other than linear, e.g., with kernels:* In a kernelized setting, one could use kernelized
Online Gradient Descent as a regression oracle, which is dimension-independent (scaling instead with the RKHS norm)
but has a $T^{\frac{1}{2}}$ regret rather than $d\log(T)$. On the other hand, one can always rely on the standard kernel online ridge
regression regret bound, that replaces bound $d\log(T)$ by the log determinant of the kernel Gram matrix of the data, and
then rely on the speed of eigenvalue decay of this matrix.
- *Misspecification in the case of universal kernels such as Gaussian:* With sufficiently small bandwidth, a universal
kernel can be realizable, i.e. $\varepsilon = 0$. Yet, choosing small bandwidth comes at a cost of increasing sample complexity,
and the optimal results for a particular problem instance may be obtained by trading off kernel bandwidth versus
misspecification.

[Meta-Review · NeurIPS 2020]

The reviewers agree that the problem addressed is significant, the results are novel, and the analysis is non-trivial. The author's response addresses most minor technical concerns raised by the reviewers. There were some suggestions made to improve the clarity of the paper that I recommend the authors to take into account while preparing the camera-ready version.